# Oxidative stress-induced phosphorylation of JIP4 regulates lysosomal positioning in coordination with TRPML1 and ALG2

Yukiko Sasazawa[1,2] (iD), Sanae Souma[3], Norihiko Furuya[2,3] (iD), Yoshiki Miura[4], Saiko Kazuno[4], Soichiro Kakuta[4], Ayami Suzuki[3], Ryota Hashimoto[4], Hiroko Hirawake-Mogi[3], Yuki Date[3,5], Masaya Imoto[2], Takashi Ueno[4], Tetsushi Kataura[6] (iD), Viktor I Korolchuk[6], Taiji Tsunemi[3], Nobutaka Hattori[1,2,3,7,*] (iD) & Shinji Saiki[2,3,**] (iD)

## Abstract

Retrograde transport of lysosomes is recognised as a critical autophagy regulator. Here, we found that acrolein, an aldehyde that is significantly elevated in Parkinson's disease patient serum, enhances autophagy by promoting lysosomal clustering around the microtubule organising centre via a newly identified JIP4-TRPML1-ALG2 pathway. Phosphorylation of JIP4 at T217 by CaMK2G in response to $Ca^{2+}$ fluxes tightly regulated this system. Increased vulnerability of JIP4 KO cells to acrolein indicated that lysosomal clustering and subsequent autophagy activation served as defence mechanisms against cytotoxicity of acrolein itself. Furthermore, the JIP4-TRPML1-ALG2 pathway was also activated by $H_2O_2$, indicating that this system acts as a broad mechanism of the oxidative stress response. Conversely, starvation-induced lysosomal retrograde transport involved both the TMEM55B-JIP4 and TRPML1-ALG2 pathways in the absence of the JIP4 phosphorylation. Therefore, the phosphorylation status of JIP4 acts as a switch that controls the signalling pathways of lysosoma l distribution depending on the type of autophagy-inducing signal.

**Keywords** autophagy; JIP4; lysosomal positioning; oxidative stress; Parkinson's disease

**Subject Categories** Autophagy & Cell Death; Membrane & Trafficking; Post-translational Modifications & Proteolysis

**The EMBO Journal (2022) 41: e111476**

## Introduction

Lysosomes play critical roles in regulating a wide range of cellular processes, including the catabolic autophagy pathway which maintains cellular homeostasis (Cabukusta & Neefjes, 2018; Ballabio & Bonifacino, 2020). Mammalian cells contain 50–1,000 lysosomes, which are located in at least two spatially distinct regions: peripheral areas close to the plasma membrane and near the microtubule-organising centre (MTOC) proximal to the nucleus. Although peripheral localisation of lysosomes plays a critical role in cancer malignancy (Nishimura *et al*, 2003; Macpherson *et al*, 2014; Marchesin *et al*, 2015; Dykes *et al*, 2016), their clustering around the MTOC contributes to upregulation of autophagy. Macroautophagy (hereafter referred to as autophagy) is a catabolic pathway degrading intracellular components via the lysosome. Autophagy ensures cellular and organismal homeostasis and ultimately survival by recycling dysfunctional proteins, protein aggregates and entire organelles, thereby preventing cellular stress (Klionsky *et al*, 2021). During autophagy, a part of the cytoplasm is engulfed by the isolation membrane to generate a double-membraned organelle termed the autophagosome. The latter subsequently fuses with a lysosome to form autolysosome, resulting in the degradation of the vesicle contents by lysosomal hydrolases. Lysosomal retrograde transport regulates autophagic flux by facilitating autophagosome formation by suppressing the mechanistic target of rapamycin complex 1 (mTORC1) and facilitating fusion between autophagosomes and lysosomes (Kimura *et al*, 2008; Korolchuk *et al*, 2011). Three molecular mechanisms that promote retrograde transport have been reported: (i) the small GTPase Rab7-Rab7 effector Rab-interacting lysosomal protein (RILP) pathway (Johansson *et al*, 2007; Rocha *et al*, 2009), (ii) transient receptor potential

1 Research Institute for Diseases of Old Age, Juntendo University Graduate School of Medicine, Tokyo, Japan
2 Division for Development of Autophagy Modulating Drugs, Juntendo University Faculty of Medicine, Tokyo, Japan
3 Department of Neurology, Juntendo University Faculty of Medicine, Tokyo, Japan
4 Biomedical Research Core Facilities, Juntendo University Graduate School of Medicine, Tokyo, Japan
5 Department of Biology, Graduate School of Science and Engineering, Chiba University, Chiba, Japan
6 Biosciences Institute, Faculty of Medical Sciences, Campus for Ageing and Vitality, Newcastle University, Newcastle upon Tyne, UK
7 Neurodegenerative Disorders Collaborative Laboratory, RIKEN Center for Brain Science, Saitama, Japan
*Corresponding author. Tel: +81 3 3813 3111; E-mail: nhattori@juntendo.ac.jp
**Corresponding author. Tel: +81 3 3813 3111; Fax: +81 3 5800 0547; E-mail: ssaiki@juntendo.ac.jp

mucolipin 1 (TRPML1) apoptosis-linked gene 2 (ALG2) pathway (Li *et al*, 2016) and (iii) lysosomal membrane protein TMEM55B-JNK-interacting protein 4 (JIP4) pathway (Willett *et al*, 2017). Recently, RUFY3 and RUFY4 were also shown to form a complex with ARL8 and JIP4-dynein complex to facilitate lysosomal retrograde transport (Keren-Kaplan *et al*, 2022; Kumar *et al*, 2022). However, the respective roles of these regulatory mechanisms as mediators of lysosomal retrograde transport in response to specific autophagy-inducing signals remain to be elucidated.

Polyamine spermidine (spd) promotes longevity via autophagy induction in neurons *in vivo* (Gupta *et al*, 2013; Minois *et al*, 2014). In addition to increased levels of acetylated spd previously reported in Parkinson's disease (PD) patients (Paik *et al*, 2010; Roede *et al*, 2013), we recently showed an alteration of general polyamine metabolism in PD, particularly the decrease in spermine (spm)/spd ratio (Saiki *et al*, 2019). In response to cellular damage, spd is metabolised to spm by spermine oxidase (SMO), and a toxic α, β-unsaturated aldehyde acrolein is generated (Igarashi & Kashiwagi, 2021). Acrolein is a key factor in perpetuating oxidative stress (Uchida *et al*, 1998; Stevens & Maier, 2008; Hamann & Shi, 2009; Park *et al*, 2014) and in PD pathogenesis (Ambaw *et al*, 2018; Acosta *et al*, 2019). On the other hand, it is also known to induce autophagy in several cell types (Jiang *et al*, 2018, 2021; Luo *et al*, 2018), which might serve as a defence mechanism against oxidative stress. However, the precise target and the role of acrolein-induced autophagy in neuronal cells have not been elucidated.

Here, we initially discovered an increase in serum levels of acrolein in PD patients. Subsequently, we demonstrated that acrolein enhances autophagy by promoting lysosomal retrograde transport via a novel mechanism, a CaMK2G-phospho-JIP4-TRPML1-ALG2 pathway.

# Results

## Acrolein is elevated in PD patient serum

α, β-Unsaturated aldehyde acrolein is cytotoxic, and aldehyde-modified proteins have been detected in nigral dopaminergic neurons of PD patients (Yoritaka *et al*, 1996). Therefore, we examined the serum levels of the acrolein-Lys adduct ($N^{\epsilon}$-(3-formyl-3,4-dehydropiperidino)-lysine; FDP-Lys) in healthy controls ($n = 22$; age: $61.4 \pm 13.9$ years) and PD patients ($n = 94$; age: $65.4 \pm 11.2$ years) by a competitive enzyme-linked immunosorbent assay. Because smoking is an external factor that increases acrolein (Raju *et al*, 2013), we only examined non-smokers. As predicted by our previous study (Saiki *et al*, 2019), acrolein-Lys adduct was significantly elevated in PD samples compared to control samples (Fig 1A). Moreover, a higher content of acrolein-Lys adduct was observed even in the early phases of PD as measured by Hoehn and Yahr (H&Y) staging, a well-established PD scale (Fig 1B and Appendix Table S1).

## Spm/acrolein treatment induces lysosomal clustering, mTORC1 inhibition and autophagosome–lysosome fusion

Acrolein induces autophagy in various cell lines (Jiang *et al*, 2018, 2021; Luo *et al*, 2018). Because alterations in lysosomal positioning regulate autophagy (Kimura *et al*, 2008; Korolchuk *et al*, 2011), we examined whether polyamines including acrolein affect the lysosomal distribution. Among polyamines, only spm, spd and *N*1-acetylspermine (*N*1-AcSpm), which were previously reported as autophagy activators (Saiki *et al*, 2019), drastically shifted the lysosomal distribution towards the perinuclear region in SH-SY5Y human neuroblastoma cells (Fig 2A). Additionally, treatment with acrolein induced lysosomal redistribution and autophagy (Figs 2B and EV1A). Acrolein is generated from spm or spd by bovine serum amine oxidase in a culture medium (Holbert *et al*, 2020), as well as by spermine oxidase in human cells (Appendix Fig S1). Both lysosomal clustering and autophagy induction by spm were cancelled by cotreatment with aminoguanidine, a diamine oxidase inhibitor that inhibits the conversion of spm to acrolein (Figs 2C and EV1B), suggesting that spm-induced lysosomal redistribution and subsequent autophagy were mediated by acrolein. Immunostaining of γ-tubulin, a MTOC marker, showed that spm/acrolein induced translocation of lysosomes to the MTOC (Figs EV1C and 2D). The lysosomal translocation was prevented by nocodazole, an inhibitor of microtubule polymerisation, and ciliobrevin D, a specific inhibitor of dynein

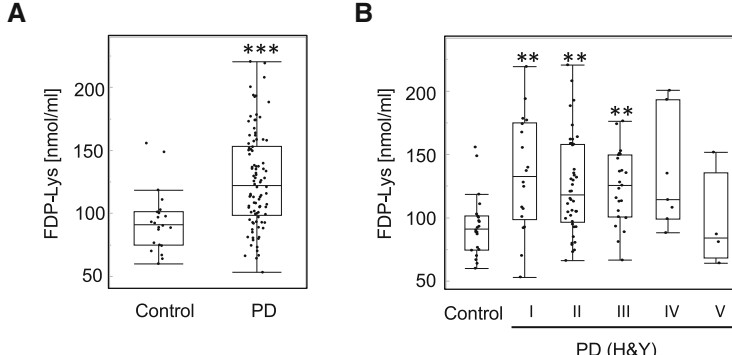

**Figure 1. Serum acrolein levels in PD patients.**

A, B The amount of FDP-lysine, which was major protein-bound acrolein, in the sera of healthy controls and PD patients was determined by an ELISA ((A) all patients (control $n = 22$; PD $n = 94$)) and (B) staged by the H&Y scale, a five-steps index of PD severity. A larger number indicates a more severe condition. $n = 94$; H&Y stage I ($n = 18$), stage II ($n = 42$), stage III ($n = 23$), stage IV ($n = 7$) and stage V ($n = 4$). The central mark in the box is the median. Box shows the 25–75 percentile range and the whiskers extend to the most extreme data points not considered outliers. \*\*\*$P < 0.0001$; \*\*$P < 0.01$. ((A) Wilcoxon test and (B) Steel's test (vs. control)).

motors (Firestone *et al*, 2012; Fig EV1D and E), suggesting that spm/acrolein promoted lysosomal retrograde transport in a dynein-dependent manner. Neither changes in proteolytic activity nor lysosomal injury by spm/acrolein were detected by Magic Red-Cathepsin B staining and the EGFP-galectin 3 accumulation assay respectively (Fig EV1F and G).

In accordance with the lysosomal distribution changes, spm, spd and acrolein suppressed mTORC1 activity as monitored by the phosphorylation status of downstream targets p70S6K and S6 (Nojima *et al*, 2003; Saiki *et al*, 2011; Figs EV2A and 2E). Additionally, spm reduced localisation of mTORC1 to lysosomes, which is required for its activity (Lawrence & Zoncu, 2019), suggesting that autophagy induction by the spm/acrolein treatment is mediated, at least in part, by the changes in mTORC1 signalling (Appendix Fig S2A and B). On the other hand, nocodazole, which inhibited lysosomal clustering, did not affect the mTORC1

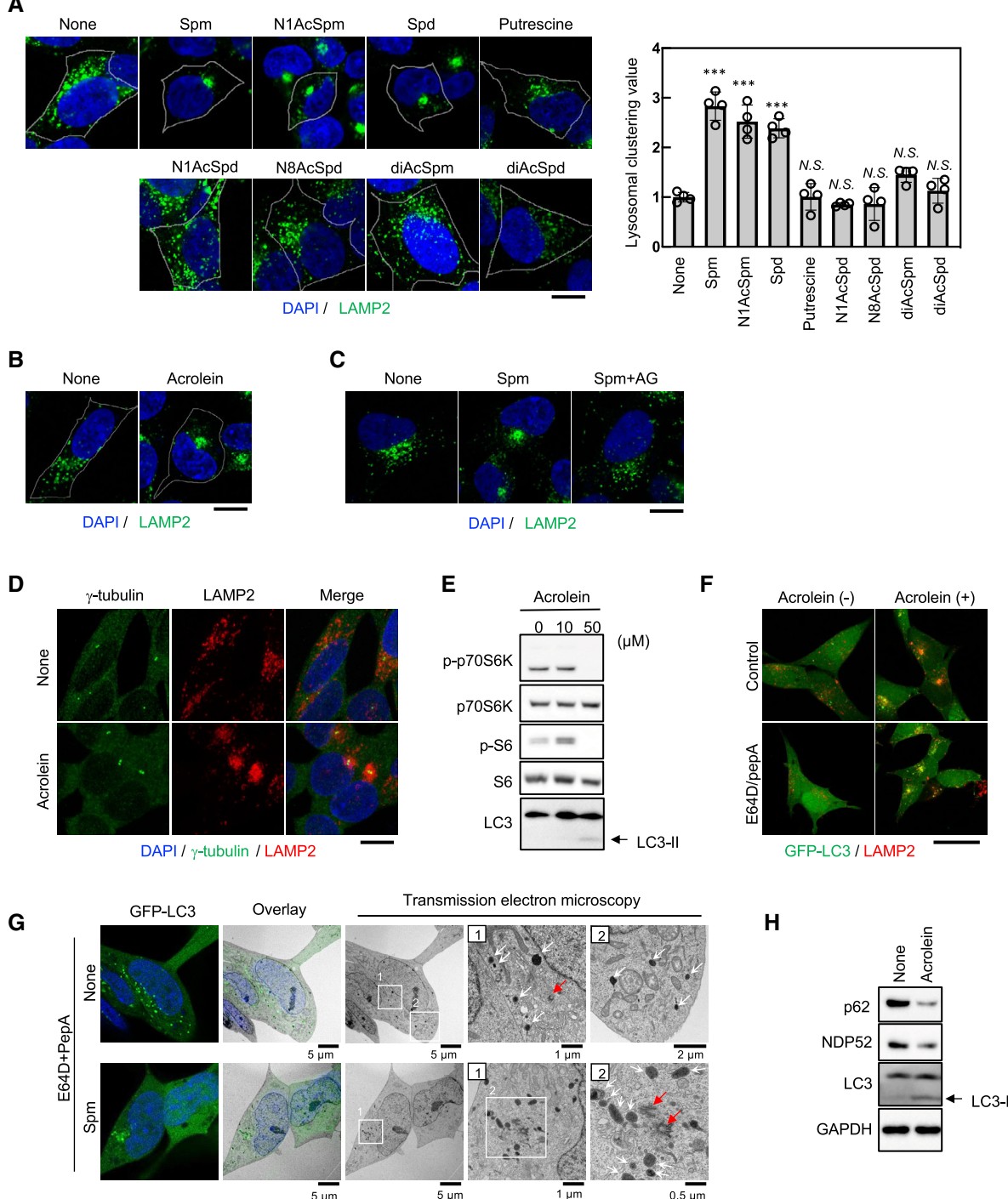

Figure 2.

**Figure 2.  Spermine metabolised to acrolein promotes lysosomal distribution towards MTOC and facilitates autophagy.**

A, B   SH-SY5Y cells were treated with the indicated chemicals at 50 μM for 4 h (A) and 40 μM acrolein for 2 h (B). Cells were fixed and stained with an anti-LAMP2 (green) antibody as a lysosomal marker. Nuclei were stained with DAPI (blue). The cell membrane is indicated by the white line. Scale bar, 10 μm. Clustered relative to whole lysosomes was quantified as lysosomal clustering value (A, right). \*\*\*$P < 0.0001$; *N.S.*, not statistically different (Dunnett's test, vs. NT). Four images were quantified per each condition. Values are mean ± SD. At least three experiments were replicated.

C   SH-SY5Y cells pre-treated with 1 mM aminoguanidine (AG) for 1 h were treated with spm for 4 h. Cells were fixed and stained with an anti-LAMP2 antibody. Scale bar, 20 μm.

D   SH-SY5Y cells were treated with 40 μM acrolein for 2 h. Cells were fixed and stained with an anti-γ-tubulin (green) antibody to detect MTOC and an anti-LAMP2 (red) antibody. Nuclei were stained with DAPI (blue). Scale bar, 20 μm.

E   SH-SY5Y cells were treated with acrolein for 2 h. Cell lysates were immunoblotted with the indicated antibodies.

F   SH-SY5Y cells stably expressing GFP-LC3 were pre-treated with 10 μg/ml E64D plus pepstatin A for 2 h and then treated with 40 μM acrolein for an additional 2 h. Cells were fixed and stained with an anti-LAMP2 (red) antibody. Nuclei were stained with DAPI (blue). Scale bar, 20 μm.

G   SH-SY5Y cells stably expressing GFP-LC3 were pre-treated with 10 μg/ml E64D plus pepstatin A for 2 h and then treated with 50 μM spm for an additional 4 h, followed by CLEM analysis. Structures containing GFP were identified by confocal microscopy in fixed cells and then relocated after Epon embedding and imaged by electron microscopy. Lysosome-like structures and centrosomes are indicated by white and red arrows respectively. Boxed regions (1, 2) are enlarged on the right.

H   SH-SY5Y cells were treated with acrolein for 2 h. Cell lysates were immunoblotted with the indicated antibodies.

Source data are available online for this figure.

inactivation by acrolein (Fig EV2B). Additionally, Torin1, a specific mTOR inhibitor, did not affect the lysosomal clustering, indicating that acrolein-induced mTORC1 inactivation is independent of lysosomal clustering (Fig EV2C).

Next, we found that colocalisation of LC3 (autophagosome marker) with LAMP2 (lysosome marker) was facilitated by spm/acrolein treatment, indicating accelerated autophagosome–lysosome fusion (autolysosome formation). This was further pronounced in the presence of E64D and pepstatin A, inhibitors of lysosomal proteolytic enzymes (Figs EV2D and 2F). The structure of spm-induced perinuclear autolysosomes was analysed by correlative light-electron microscopy (CLEM) in the presence of E64D and pepstatin A. SH-SY5Y cells stably expressing GFP-LC3 and treated with spm exhibited clustered lysosome-like vesicles in the perinuclear region, particularly near the centrosome, where GFP-LC3 was detected (Fig 2G). Enhancement of autophagic flux was further supported by increased degradation of autophagy receptor proteins and substrates NDP52 and p62 (Mizushima & Levine, 2020; Figs 2H and EV2E). These data suggest that spm via its metabolite acrolein inactivates mTORC1, which contributes to the upregulation of autophagosome biogenesis, and also enhances lysosomal retrograde transport which results in an increase in autolysosome formation without an overt effect on the lysosomal structure or enzymic activity. These two independent effects synergistically contribute to promoting effective autophagy.

## Spm/acrolein affects the lysosomal distribution through Ca²⁺-TRPLM1-ALG2-dynein signalling

Three dynein-mediated pathways for lysosomal retrograde movement have been reported (Rab7-RILP (Johansson et al, 2007; Rocha et al, 2009), TRPML1-ALG2 (Li et al, 2016) and TMEM55B- JIP4 (Willett et al, 2017) pathways). Therefore, we examined whether any of these mechanisms are involved in the changes in lysosomal distribution induced by spm/acrolein. To this end, we initially explored the effects of Rab7, ALG2, TRPML1, TMEM55B and JIP4 knockdown. Spm-induced lysosomal perinuclear clustering was observed in control and Rab7-silenced cells (Fig 3A), but it was almost completely cancelled by the knockdown of ALG2-, TRPML1 or JIP4 (Fig 3B and C). Knockdown of TMEM55B only partially suppressed lysosomal clustering (Fig 3C). Similar results were obtained

when acrolein was used instead of spm (Fig 3D). On the other hand, TMEM55A, which exhibits a high degree of homology with TMEM55B but lacks JIP4-binding site (Willett et al, 2017), did not affect the acrolein-induced lysosomal movement (Appendix Fig S3A). Knockdown efficiency of these genes was confirmed by western blotting and quantitative reverse transcription–PCR (qRT–PCR; Appendix Fig S3B–E). These data indicate that spm/acrolein-induced retrograde transport of lysosomes is primarily mediated by the TRPML1-ALG2 pathway.

TRPML1 is a lysosomal Ca²⁺ channel regulated by binding to its endogenous agonist phosphatidylinositol-3,5-bisphosphate (PI(3,5)P₂) produced by PIKfyve from PI(3)P (Dong et al, 2010). When Ca²⁺ is released from the lysosome by TRPML1, ALG2, an EF-hand-containing protein, associates physically with TRPML1 (Vergarajauregui et al, 2009) and the minus-end-directed dynein motor, to enhance lysosomal retrograde movement (Li et al, 2016). Consistent with this mechanism, spm-induced lysosomal retrograde transport was significantly inhibited by the treatment with BAPTA-AM, a Ca²⁺ chelator, or YM201636, a PIKfyve inhibitor (Jefferies et al, 2008; Appendix Fig S3F). Moreover, live cell calcium imaging by Fura2-AM demonstrated that acrolein treatment induced repetitive Ca²⁺ spikes (Fig. 3E and Movie EV1). Additionally, both the frequency and amplitude of the spikes were reduced by TRPML1 knockdown (Fig 3F and G, and Movies EV2 and EV3). These findings indicated that Ca²⁺ released by TRPML1 on lysosomes in response to spm/acrolein caused their retrograde transport through activation of the TRPML1-ALG2-dynein pathway.

## Phosphorylated JIP4 is involved in spm/acrolein-induced lysosomal movement

Independent of TRPML1-ALG2-dynein pathway, JIP4 is known to regulate lysosomal retrograde movement by binding to TMEM55B (Willett et al, 2017; Ballabio & Bonifacino, 2020). However, our results demonstrated that silencing of JIP4, but not TMEM55B, almost completely suppressed the effect of spm/acrolein on lysosomal distribution (Fig 3C and D), suggesting that JIP4 contributed to the spm/acrolein-induced retrograde transport in a TMEM55B-independent manner. Indeed, immunoprecipitation assays showed that the JIP4 and TMEM55B interaction was not enhanced by spm or acrolein (Figs EV3A and 4A). Although JIP3 is also known to be

associated with lysosomal retrograde transport (Gowrishankar *et al*, 2017), JIP3 as well as other JIP family proteins JIP1 and JIP2 were not involved in the spm-induced lysosomal clustering (Appendix Fig S4A). Furthermore, JIP4 accumulated near perinuclear lysosomal clusters induced by spm/acrolein (Figs EV3B and

4B). These data suggested that spm/acrolein induced JIP4 translocation to the lysosomal surface by an unknown mechanism, thereby promoting changes in the lysosomal distribution. Moreover, both acrolein-induced lysosomal clustering and JIP4 relocalisation were blocked by the cotreatment with an antioxidant *N*-acetyl-L-cysteine,

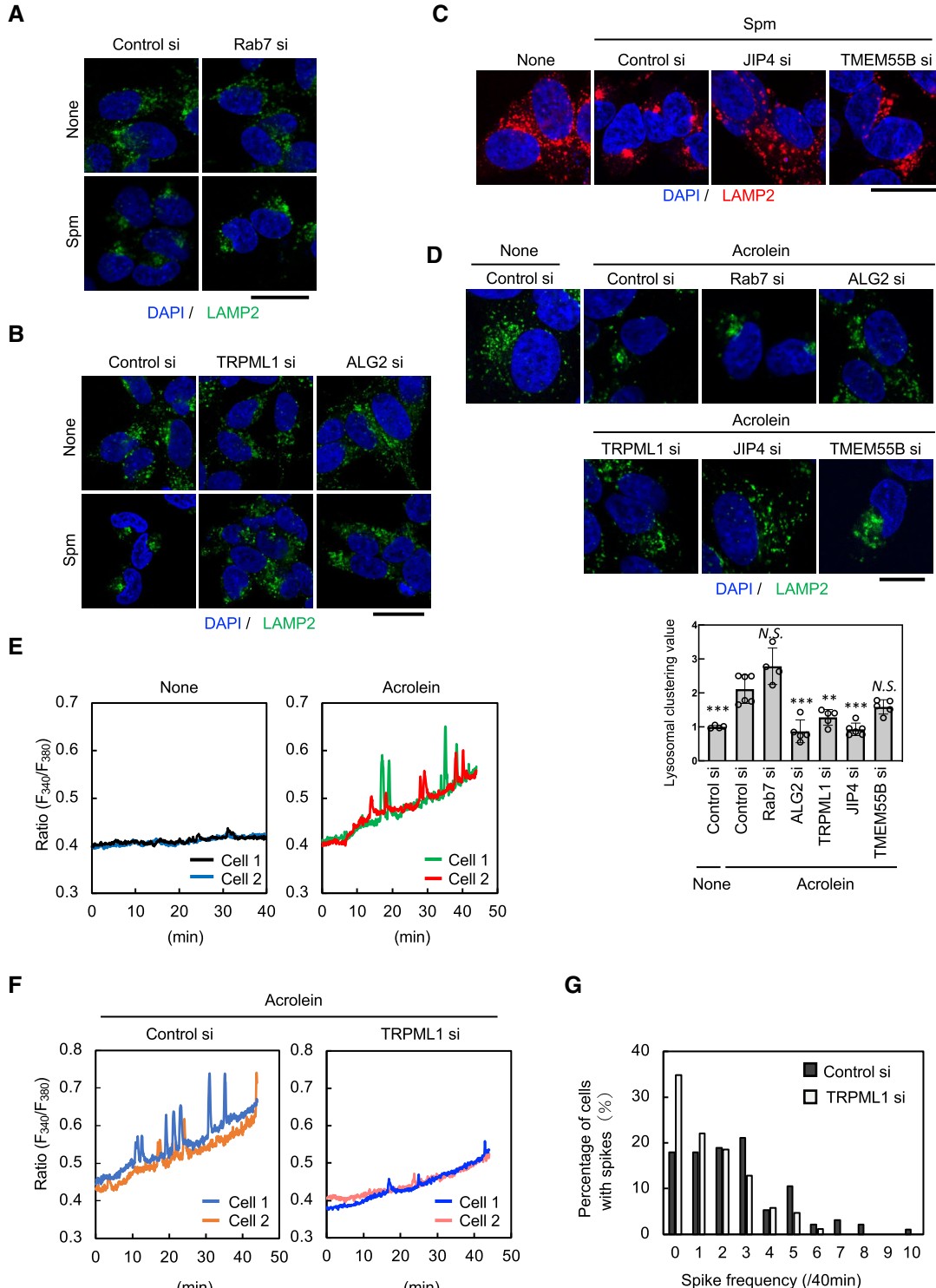

**Figure 3.**

**Figure 3. Spm/acrolein induces perinuclear lysosomal clustering through the TRPML1-ALG2-dynein pathway.**

A, B   SH-SY5Y cells were transfected with the indicated siRNA for 48 h and then treated with 50 μM spm for 4 h. Cells were fixed and stained with an anti-LAMP2 antibody (green). Nuclei were stained with DAPI (blue). Scale bar, 20 μm.

C   SH-SY5Y cells were transfected with JIP4 siRNA or TMEM55B siRNA for 48 h, and then treated with 50 μM spm for 4 h. Cells were fixed and stained with an anti-LAMP2 antibody (red). Nuclei were stained with DAPI (blue). Scale bar, 20 μm.

D   SH-SY5Y cells were transfected with the indicated siRNAs for 48 h and then treated with 40 μM acrolein for 2 h. (Upper) Cells were fixed and stained with an anti-LAMP2 antibody (green). Nuclei were stained with DAPI (blue). Scale bar, 10 μm. (Lower) Clustered relative to whole lysosomes was quantified as lysosomal clustering value. ***$P < 0.0001$; **$P < 0.001$; *$P < 0.01$; N.S., not statistically different (Dunnett's test, vs. control si-acrolein). Four to six images were quantified per each condition. Values are mean ± SD. At least three experiments were replicated.

E   Fura2-AM-loaded SH-SY5Y cells were treated with 40 μM acrolein and observed by time-lapse fluorescence microscopy at 5 s intervals. The fluorescence ratio ($F_{340}/F_{380}$) values in two representative cells are shown.

F   Cells transfected with control siRNA (left) and TRPML1 siRNA (right) for 48 h were loaded with Fura2-AM for 30 min. Cells were treated with 40 μM acrolein and observed by time-lapse fluorescence microscopy at 5 s intervals. The fluorescence ratio ($F_{340}/F_{380}$) values in two representative cells are shown.

G   Histogram of the spike frequency. Spikes with an amplitude of more than 0.05 for 40 min were counted. The value represents the percentage of cells with the indicated number of spikes to the total cells. Spike frequency "0" means no spikes in the cell.

Source data are available online for this figure.

indicating the critical role of oxidative stress in this phenomenon (Fig EV3C).

Interestingly, we noticed that spm/acrolein induced a mild mobility shift of the JIP4 band in western blots with an anti-JIP4 antibody, which raised the possibility of JIP4 phosphorylation. Phos-tag PAGE, which detects a mobility shift of phosphorylated proteins, revealed that spm/acrolein induced a significant decrease in the mobility of JIP4 band (Figs EV3D and 4C). Both spm-treated and non-treated control samples showed an increase in the JIP4 mobility in the SDS gel to the same position after treatment of the cell lysates with lambda phosphatase (Fig EV3E). This result indicated that JIP4, which was phosphorylated under basal conditions, was further phosphorylated in response to spm treatment. Moreover, the spm-induced mobility shift of JIP4 was not detected in HeLa cells in which spm did not affect the lysosomal distribution (Appendix Fig S4B and C). Thus, we hypothesised that spm/acrolein-induced phosphorylation of JIP4 played a role in the lysosomal distribution phenotype. To test this hypothesis, we first identified the kinase responsible for phosphorylating JIP4 in response to spm/acrolein using a protein kinase inhibitor library. Among approximately 100 kinase inhibitors investigated, JAK3 inhibitor VI and PKC inhibitor Gö6976 suppressed the changes in the spm-induced lysosomal distribution (Fig EV3F and G). Furthermore, Jak3 inhibitor VI clearly suppressed both the JIP4 phosphorylation induced by spm/acrolein (Figs EV3H and 4D) and the changes in lysosomal distribution and JIP4 translocation to perinuclear lysosomes (Figs EV3F and 4E). However, knockdown of Jak3 or an alternative PKC inhibitor Gö7874 did not affect spm-induced lysosomal distribution, suggesting that the above inhibitors exerted these effects by suppressing a protein kinase other than the original target (Fig EV3I and J). By profiling the activity of 178 commercially available kinase inhibitors against a panel of 300 recombinant protein kinases, a previous report revealed the identity of off-target effects (Davis *et al*, 2011). By cross-referencing this profiling, we selected Ca$^{2+}$/calmodulin-dependent protein kinase II (CaMK2) as a potential kinase responsible for JIP4 phosphorylation because CaMK2 is potently inhibited by JAK3 inhibitor VI and Gö6976, both of which attenuated the spm-induced effects. CaMK2 is a Ca$^{2+}$-dependent kinase auto-phosphorylated and activated in response to Ca$^{2+}$ influx (Braun & Schulman, 1995; Coultrap & Bayer, 2012). In the CaMK2 family, only siRNA against CaMK2G completely prevented spm/acrolein-induced JIP4 phosphorylation (Figs EV3K and 4F), changes

in the lysosomal localisation of JIP4 and the lysosomal retrograde transport (Figs EV3L and 4G). Besides, CaMK2G knockdown did not alter Ca$^{2+}$ release, indicating that CaMK2G acts downstream of TRPML1 (Appendix Fig S4D). Collectively, our data suggested that Ca$^{2+}$ release through TRPML1 in response to acrolein (Fig 3F and G) activates CaMK2G that, in turn, phosphorylates JIP4.

### Phosphorylation of JIP4 at T217 is important to control lysosomal positioning

To determine the phosphorylation sites of JIP4 responsible for the spm/acrolein-induced lysosomal distribution change, myc-DDK (flag)-tagged JIP4-overexpressing SH-SY5Y cells were treated with acrolein in the presence or absence of Jak3 inhibitor VI. JIP4 was immunoprecipitated from cell lysates using anti-flag magnetic beads and eluted with the flag peptide (Appendix Fig S5A). The samples were digested with trypsin, and the peptides were analysed by sequential window acquisition of all theoretical fragment ion spectra mass spectrometry (SWATH-MS), resulting in the detection of five phosphorylated peptides derived from JIP4 (Fig 4H). The number of phospho-peptides containing T217 was significantly increased by acrolein treatment and decreased by cotreatment with Jak3 inhibitor VI (Fig 4I), indicating that T217 was the phosphorylation site of JIP4 by CaMK2G upon spm/acrolein treatment. To determine whether CaMK2G directly phosphorylates JIP4, we further performed an *in vitro* kinase assay. His$_6$-tagged recombinant truncated 100–300 aa JIP4 was purified from *Escherichia coli* (Appendix Fig S5B) and reacted with recombinant CaMK2G with/without Ca$^{2+}$ and calmodulin. CaMK2G strongly phosphorylated JIP4 100–300 aa polypeptide in a Ca$^{2+}$/calmodulin-dependent manner, which was effectively suppressed by Jak3 inhibitor VI (Fig 4J).

Next, we examined whether JIP4 phosphorylation at T217 was involved in the lysosomal redistribution induced by spm/acrolein. JIP4 knockout (KO) SH-SY5Y cells were generated using CRISPR-Cas9 technology (Appendix Fig S6A and B) followed by re-expression of flag-tagged wild-type or phosphorylation-defective (T217A) JIP4. In JIP4 KO cells, acrolein completely lost its ability to affect lysosomal distribution. Expression of wild-type flag-JIP4, but not the JIP4 phosphorylation-defective mutant (flag-JIP4 T217A), rescued the lysosomal clustering phenotype (Fig 4K), indicating that phosphorylation of JIP4 at T217 was indispensable for the acrolein-induced lysosomal distribution change.

## JIP4 regulates lysosomal positioning in coordination with TRPML1 and ALG2

Next, we determined how JIP4 contributed to TRPML1-ALG2-dependent lysosomal retrograde transport induced by spm/acrolein.

Acrolein treatment induced lysosomal recruitment of JIP4, which was suppressed by silencing TRPML1 or ALG2, but not TMEM55B (Fig 5A). Acrolein-induced lysosomal localisation of JIP4 was further confirmed by another method, a lysosomal immunoprecipitation (Lyso-IP) technique enabling rapid isolation of intact lysosomes

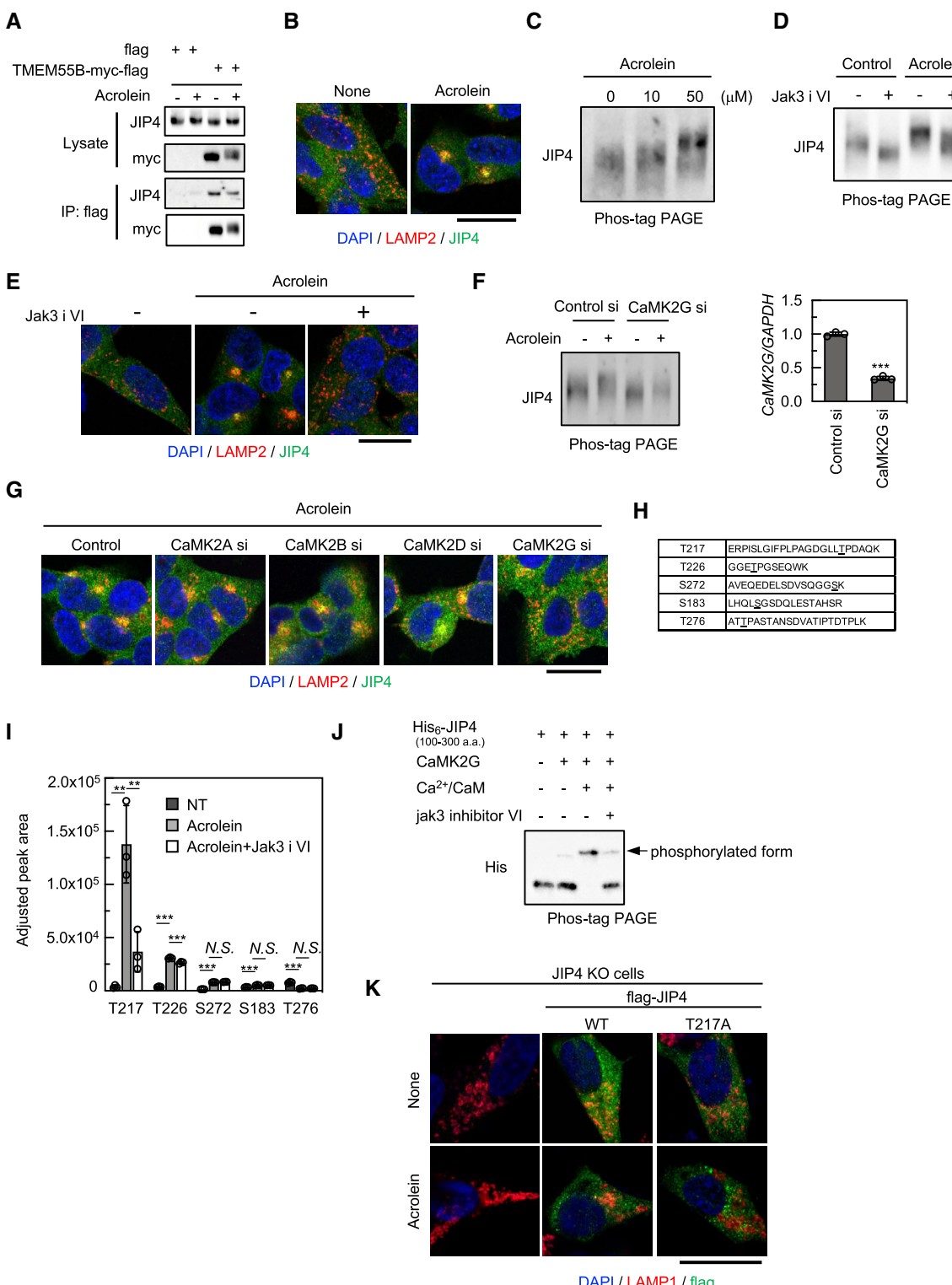

**Figure 4.**

◀

**Figure 4.   Phospho-JIP4 mediates acrolein-induced lysosomal clustering.**

A    Cells were transfected with the TMEM55B-myc-DDK plasmid for 24 h and then treated with 40 μM acrolein for 2 h. Cell lysates were immunoprecipitated with flag magnetic beads and immunoblotted with the indicated antibodies.

B    SH-SY5Y cells were treated with 40 μM acrolein for 2 h. Cells were fixed and stained with anti-JIP4 (green) and anti-LAMP2 (red) antibodies. Nuclei were stained with DAPI (blue). Scale bar, 20 μm.

C    SH-SY5Y cells treated with the indicated concentration of acrolein for 2 h were lysed, subjected to Phos-tag PAGE and immunoblotted with an anti-JIP4 antibody.

D    SH-SY5Y cells were pre-treated with 10 μM Jak3 inhibitor VI (Jak3 i VI) for 1 h and then treated with 40 μM acrolein for an additional 2 h. Cell lysates were subjected to Phos-tag PAGE and immunoblotted with an anti-JIP4 antibody.

E    SH-SY5Y cells were pre-treated with 10 μM Jak3 inhibitor VI for 1 h and then treated with 40 μM acrolein for an additional 2 h. Cells were fixed and stained with anti-JIP4 (green) and anti-LAMP2 (red) antibodies. Nuclei were stained with DAPI (blue). Scale bar, 20 μm.

F, G  SH-SY5Y cells transfected with indicated siRNAs for 48 h were treated with 40 μM acrolein for 2 h. Cell lysates were subjected to Phos-tag PAGE and immunoblotted with anti-JIP4 antibody (F, left). CaMK2G knockdown efficiency was confirmed by qRT–PCR; ***$P < 0.001$, $n = 3$ technical replicates. Values are mean ± SD. At least three experiments were replicated (F, right). Cells were fixed and stained with anti-LAMP2 and anti-JIP4 antibodies. Scale bar, 20 μm (G).

H    Peptide sequence detected by mass spectrometry. The underlined amino acid was modified with phosphoric acid.

I    The amount of each phosphorylated peptide when cells were treated with 40 μM acrolein or cotreated with acrolein and JAK3 inhibitor VI compared with the control was determined by SWATH-MS analysis. ***$P < 0.001$, **$P < 0.01$; *N.S.*, not statistically different. (Tukey–Kramer's test); $n = 3$ technical replicates. Values are mean ± SD.

J    His$_6$-tagged recombinant truncated JIP4 (100–300 aa) was purified from *E. coli* and incubated with recombinant CaMK2G with/without Ca$^{2+}$ and calmodulin at 30°C for 1 h. The reactant was subjected to Phos-tag PAGE and immunoblotted with anti-His antibody. Jak3 inhibitor VI was pre-reacted with CaMK2G for 30 min.

K    JIP4 KO cells were transfected with flag-tagged JIP4 (WT and T217A) for 24 h and treated with 40 μM acrolein for 2 h. Cells were fixed and stained with the indicated antibodies. Scale bar, 20 μM.

Source data are available online for this figure.

(Abu-Remaileh *et al*, 2017; Fig EV4). Moreover, TRPML1–ALG2 interaction in response to acrolein was confirmed by immunocytochemistry and immunoprecipitation assays (Fig 5B and C). These results suggested that the TRPML1-ALG2 complex played a role in retaining the motor adaptor JIP4 on the lysosomal surface, thereby promoting dynein-dependent retrograde transport of the lysosome. Indeed, partial colocalisation of JIP4 with TRPML1 was observed in acrolein-treated SH-SY5Y cells stably expressing TRPML1-mCherry (Fig 5D). Moreover, transiently expressed GFP-ALG2 was found near JIP4 with partial colocalisation detectable upon acrolein treatment (Fig 5E). A physical interaction between GFP-ALG2 and endogenous JIP4 was confirmed by a proximity ligation assay (PLA). Importantly, acrolein treatment enhanced the interaction of GFP-ALG2 with JIP4 (Fig 5F). Considering that CaMK2G phosphorylates JIP4 in response to acrolein treatment, this phosphorylation is required for the association of JIP4 with the TRPML1-ALG2 complex to promote retrograde transport of lysosomes.

### Lysosomal clustering in response to H$_2$O$_2$ or acrolein is dependent on phospho-JIP4

As highlighted above, perinuclear localisation of lysosomes plays an important role in autophagy. Indeed, acrolein did not increase autophagic flux in JIP4 KO cells compared with wild-type SH-SY5Y cells (Fig 6A). Moreover, both JAK3 inhibitor VI and Ca$^{2+}$ chelator BAPTA-AM suppressed acrolein-induced autophagic flux (Fig 6B), indicating that lysosomal retrograde transport mediated by phospho-JIP4-TRPML1-ALG2 pathway was a critical molecular mechanism of acrolein-induced autophagy.

Next, we investigated whether this pathway was also required for autophagy induction by other stimuli. Both oxidative stress (H$_2$O$_2$ or acrolein treatment) and nutrient starvation-induced lysosomal retrograde transport in parental, but not in JIP4 KO SH-SY5Y cells, indicating the requirement for JIP4 in lysosomal retrograde transport induced by these conditions (Fig 6C). However, BAPTA-AM or Jak3 inhibitor VI suppressed the lysosomal retrograde transport induced by acrolein or H$_2$O$_2$, but not starvation (Fig 6D).

Moreover, H$_2$O$_2$ induced JIP4 phosphorylation, but starvation did not (Fig 6E). These observations suggested that H$_2$O$_2$ and acrolein, but not starvation, induced lysosomal retrograde transport in a phospho-JIP4 (T217)-dependent manner. Conversely, knockdown of TMEM55B clearly suppressed lysosomal retrograde transport induced by starvation, whereas lysosomal clustering induced by H$_2$O$_2$ or acrolein was only partially inhibited. On the other hand, both TRPML1 and ALG2 depletion suppressed lysosomal clustering induced by not only acrolein and H$_2$O$_2$ but also starvation (Fig 6F, Appendix Fig S6C). These data suggest that oxidative stress triggered by the acrolein or H$_2$O$_2$ treatment facilitates lysosomal retrograde transport via the phospho-JIP4-TRPML1-ALG2 pathway, whereas starvation appeared to drive lysosomal retrograde transport primarily via the TRPML1-ALG2 and TMEM55B-JIP4 pathways in the absence of JIP4 phosphorylation (Fig 7A).

Finally, we examined the effect of perinuclear lysosomal localisation on acrolein-induced cytotoxicity. JIP4 KO SH-SY5Y cells as well as TRPML1 knock-downed cells were more vulnerable to acrolein treatment compared with parental cells on the basis of changes in cell morphology and an LDH assay (Fig 7B and C and Appendix Fig S6D), suggesting that JIP4-mediated lysosomal clustering and subsequent autophagy exerted a cytoprotective effect against acrolein toxicity. Together, our findings demonstrate that regulation of lysosomal redistribution by the JIP4-TRPML1-ALG2 system controls a broad cellular stress response via autophagy enhancement.

## Discussion

In this study, we found that the amount of acrolein in PD patients was high even at the early stages of the disease compared with healthy controls, which likely results from the alteration of polyamine metabolism in PD patients (Saiki *et al*, 2019). Moreover, we demonstrated that acrolein induces autophagy by accelerating lysosomal retrograde transport via a novel mechanism, a TRPML1-ALG2-phospho-JIP4 pathway. Previous *in vitro* and *in vivo* studies

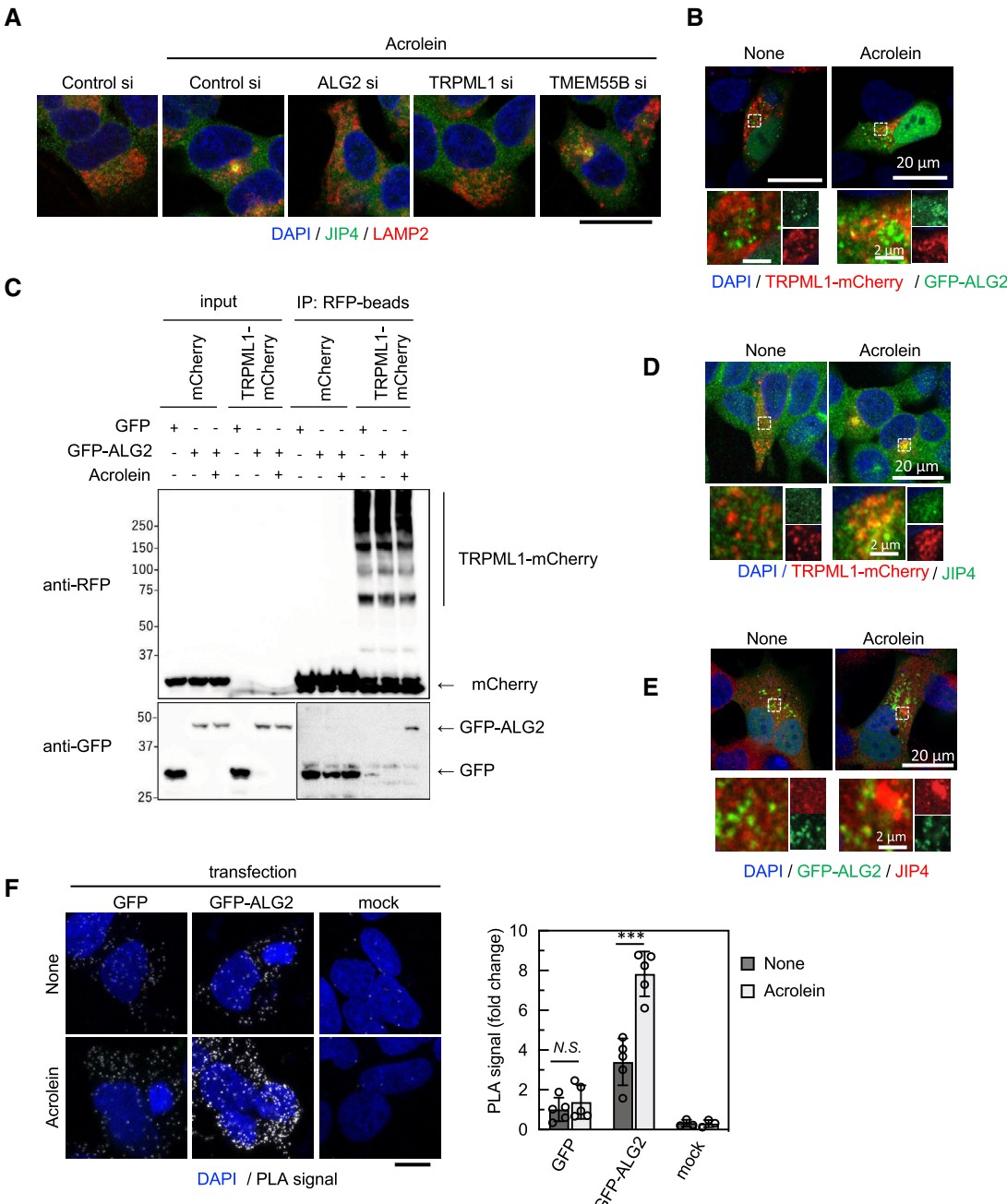

**Figure 5. JIP4-dependent lysosomal retrograde transport requires TRPML1-ALG2 signalling.**

A   SH-SY5Y cells were transfected with the indicated siRNAs for 48 h and then treated with 40 μM acrolein for 2 h. Cells were fixed and stained with anti-JIP4 (green) and anti-LAMP2 (red) antibodies. Nuclei were stained with DAPI (blue). Scale bar, 20 μm.

B   TRPML1-mCherry stably expressing SH-SY5Y cells were transfected with GFP-ALG2 for 48 h and then treated with 40 μM acrolein for 2 h. Cells were fixed, and then green and red fluorescence were detected by confocal microscopy.

C   GFP or GFP-ALG2-transfected TRPML1-mCherry stably expressing cells were treated with acrolein for 2 h. Cell lysates were immunoprecipitated with anti-RFP magnetic beads and immunoblotted with the indicated antibodies.

D   TRPML1-mCherry stably expressing SH-SY5Y cells were treated with 40 μM acrolein for 2 h. Cells were fixed and stained with an anti-JIP4 (green) antibody.

E   SH-SY5Y cells were transfected with GFP-ALG2 for 48 h and then treated with 40 μM acrolein for 2 h. Cells were fixed and stained with an anti-JIP4 (red) antibody.

F   SH-SY5Y cells transfected with GFP or GFP-ALG2 for 24 h were treated with acrolein for 2 h and then spatial proximity of JIP4 and GFP was detected by a PLA using JIP4 and GFP antibodies. Scale bar, 10 μm. ***$P < 0.001$; *N.S.*, not statistically different (Tukey–Kramer's test). Three to five images were quantified per each condition. Values are mean ± SD. At least three experiments were replicated.

Source data are available online for this figure.

have highlighted neurotoxic effects of acrolein mediated by oxidative stress and enhanced α-synuclein aggregation (Liu-Snyder et al, 2006; Wang et al, 2017; Ambaw et al, 2018; Acosta et al, 2019). In addition to caloric restriction and exercise, various phytochemicals, such as resveratrol, quercetin and spermidine, have hormetic effects on health and longevity via autophagy induction

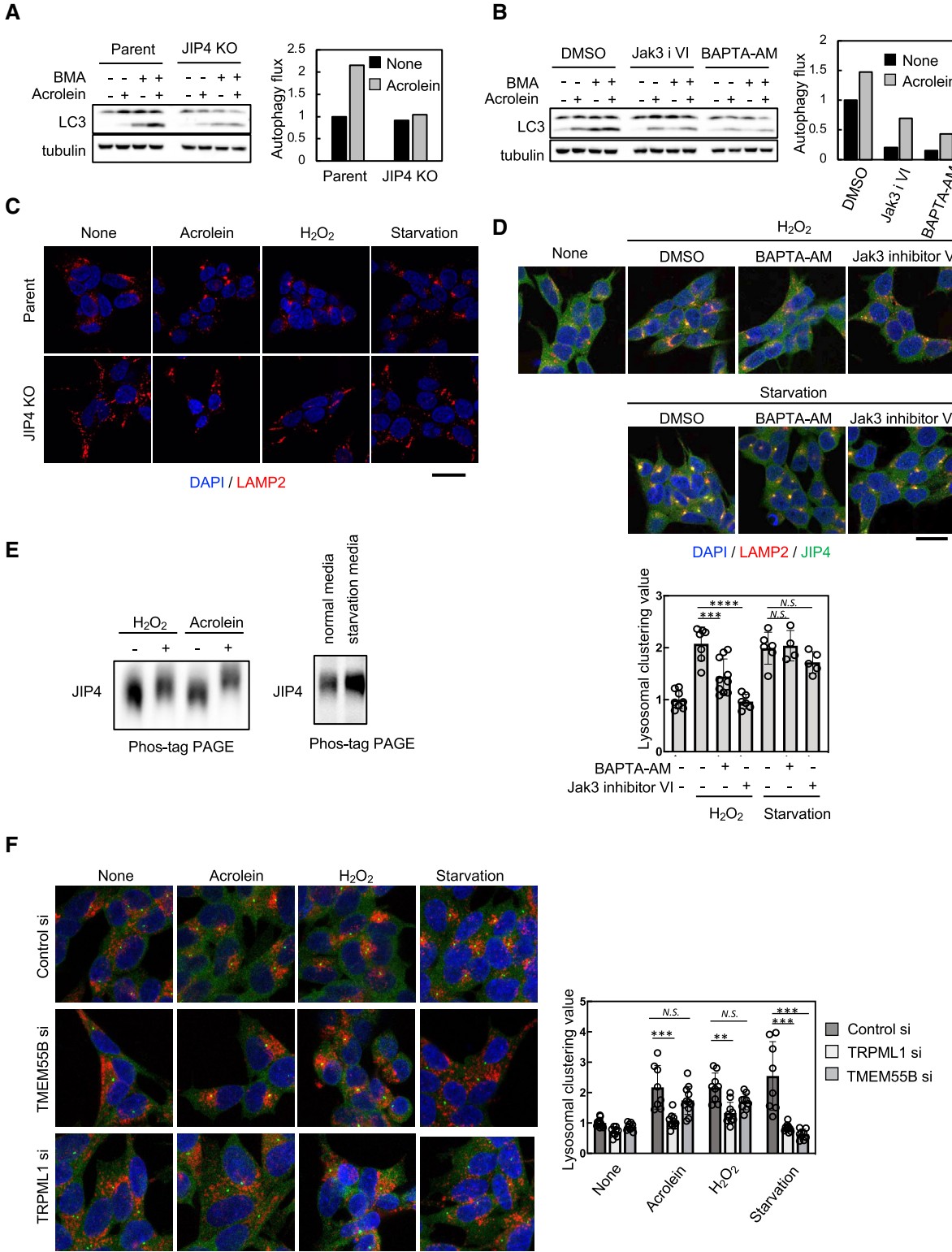

**Figure 6.**

**Figure 6. The phosphorylation state of JIP4 acts as the switch that controls lysosomal positioning.**

A, B  Parental and JIP4 KO SH-SY5Y cells were treated with 40 μM acrolein with or without 100 nM bafilomycin A1 for 2 h (A). SH-SY5Y cells were treated with 40 μM acrolein with or without 100 nM bafilomycin A1 in the presence of 10 μM Jak3 inhibitor VI (Jak3 i VI) for 2 h (B). Cell lysates were immunoblotted with the indicated antibodies. Autophagic flux was determined as the LC3-II production rate calculated as the level of LC3-II with bafilomycin A1 treatment.

C  Parental and JIP4 KO SH-SY5Y cells were treated with 40 μM acrolein and 500 μM $H_2O_2$ or cultured in the starvation medium for 2 h and then stained with an anti-LAMP2 antibody (red). Nuclei were stained with DAPI (blue). Scale bar, 20 μm.

D  (Upper) SH-SY5Y cells were pre-treated with 10 μM BAPTA-AM or 10 μM Jak3 inhibitor VI for 1 h and then treated with 500 μM $H_2O_2$ for an additional 2 h or cultured in the starvation medium for an additional 2 h. Cells were fixed and stained with anti-JIP4 (green) and anti-LAMP2 (red) antibodies. Nuclei were stained with DAPI (blue). Scale bar, 20 μm. (Lower) Clustered relative to whole lysosomes was quantified as lysosomal clustering value. ****$P$ < 0.0001, ***$P$ < 0.001; N.S., not statistically different (Tukey–Kramer's test). Four to ten images were quantified per each condition. Values are mean ± SD. At least three experiments were replicated.

E  SH-SY5Y cells were treated with 40 μM acrolein and 500 μM $H_2O_2$ or cultured in the starvation medium for 2 h. Cells were lysed, subjected to Phos-tag PAGE and immunoblotted with anti-JIP4 antibody.

F  (left) SH-SY5Y cells were transfected with TMEM55B or TRPML1 siRNAs for 48 h and then treated with 40 μM acrolein and 500 μM $H_2O_2$ or cultured in the starvation medium for 2 h. Cells were fixed and stained with anti-γ-tubulin (green) and anti-LAMP2 (red) antibodies. Nuclei were stained with DAPI (blue). Scale bar, 20 μm. (Right) Clustered relative to whole lysosomes was quantified as lysosomal clustering value. ***$P$ < 0.001, **$P$ < 0.01; N.S., not statistically different (Tukey–Kramer's test). Eight to 12 images were quantified per each condition. Values are mean ± SD. At least three experiments were replicated.

Source data are available online for this figure.

(Martel et al, 2019). Considering several lines of evidence indicating that autophagy perturbation causes neurodegeneration (Mizushima & Levine, 2020) and our results showing that JIP4 KO cells are more vulnerable to acrolein treatment compared with parental cells, acrolein might act as a hormetic to enhance the defensive response via autophagy induction at the early stages of PD despite the high toxicity of this metabolite.

Autophagy activation by acrolein or its precursor polyamines (spm, spd and N1-AcSpm) was facilitated by perinuclear clustering of lysosomes, which contributed to the enhancement of autophagic flux. We found that the spm/acrolein-induced lysosomal distribution change was regulated by TRPML1-ALG2-dynein signalling. This signalling pathway was triggered by $Ca^{2+}$ release mediated by TRPML1. Acrolein evokes $Ca^{2+}$ through TRPA1, an excitatory ion channel (Bautista et al, 2006). However, we did not detect TRPA1 expression in SH-SY5Y cells, which is consistent with gene expression databases. Thus, we concluded that TRPA1 is not related to acrolein-induced $Ca^{2+}$ influx in SH-SY5Y cells. It has also been reported that TRPML1 acts as a ROS sensor on lysosomes and orchestrates an autophagy-dependent negative-feedback programme to mitigate oxidative stress in the cell (Zhang et al, 2016). Therefore, the $PtdIns(3,5)P_2$-TRPML1-ALG2-dynein pathway can be considered to be an on-demand lysosomal retrograde transport system in response to oxidative stressors such as acrolein and $H_2O_2$.

$PtdIns(3,5)P_2$-TRPML1-ALG2 (Li et al, 2016) and TMEM55B-JIP4 pathways (Willett et al, 2017) appear to independently regulate lysosomal retrograde transport. Nevertheless, JIP4 is required for spm/acrolein-induced lysosomal retrograde transport in a TMEM55B-independent manner. The issue is how these two independent pathways are controlled separately by JIP4. Considering the qRT–PCR data showing no significant changes in mRNA expression levels of both TRPML1 and TMEM55B following 2 h treatment with acrolein, $H_2O_2$ and starvation condition (Appendix Fig S7A), we propose that the phosphorylation state of JIP4 at T217 by CaMK2G acts as a switch by altering the affinity to its binding partners to determine which pathway JIP4 uses. JIP3 is also a motor adaptor protein (Gowrishankar et al, 2017). However, JIP3 was not involved in spm/acrolein-induced lysosomal retrograde transport, possibly because JIP3 does not have an amino acid residue

homologous to T217 in JIP4. Moreover, T217 in JIP4 is conserved among vertebrate species including Homo sapiens, rat, mouse, zebrafish and chicken (Fig EV5A). These data support our conclusions regarding the importance of JIP4 phosphorylation at T217 for spm/acrolein-induced lysosomal redistribution mediated by the TRPML1-ALG2 complex. ALG2 mainly localises to ER exit sites, cytosol and cytoplasmic vesicles (Yamasaki et al, 2006), and some of them translocate to lysosomes and are involved in the regulation of lysosomal distribution (Li et al, 2016). Indeed, GFP-ALG2 only partially colocalises with lysosomes in response to acrolein as shown in Fig 5B. Acrolein-induced JIP4–ALG2 interaction may occur not only on lysosomes but also at other locations such as ER exit sites potentially explaining the lack of clustering of JIP4-ALG2 PLA signal (Fig 5E). Based on these observations, we speculate that JIP4-ALG2 complex translocated to lysosome may participate in lysosomal retrograde transport by binding to TRPML1. However, because direct interaction between phosphorylated JIP4 and the TRPML1-ALG2 complex was not detected by immunoprecipitation assays, phosphorylated JIP4 may interact with TRPML1 or ALG2 indirectly. Recently, RUFY3 was found to interact with ARL8 and JIP4-dynein complex (Keren-Kaplan et al, 2022; Kumar et al, 2022), however, our knockdown analyses revealed that RUFY3 was not involved in acrolein-induced lysosomal clustering. This result indicated that RUFY3 is not the molecule linking JIP4 to TRPML1 or ALG2 during oxidative stress (Fig EV5B). RUFY3 could be a possible adaptor between non-phosphorylated JIP4 and TRPML1 or ALG2 under starvation conditions because RUFY3 was reported to suppress starvation-induced lysosomal clustering (Kumar et al, 2022). Although the precise molecular mechanisms of lysosomal retrograde transport in response to starvation remain to be elucidated further, our study established an indispensable role of phosphorylated JIP4 (T217) in oxidative stress conditions. Multiple combinations of dynein adaptors/regulators including SEPT9 (Kesisova et al, 2021), SNAPIN (Cai et al, 2010; Yuzaki, 2010) and RUFY3/4 may allow the lysosomes to move retrogradely in response to various physiological stresses. Future identification of the molecule(s) that recruit JIP4 to the lysosomal TRPML1-ALG2 complex in various physiological stresses will help to decipher the complexity of the molecular mechanisms controlling lysosomal positioning and regulating autophagy.

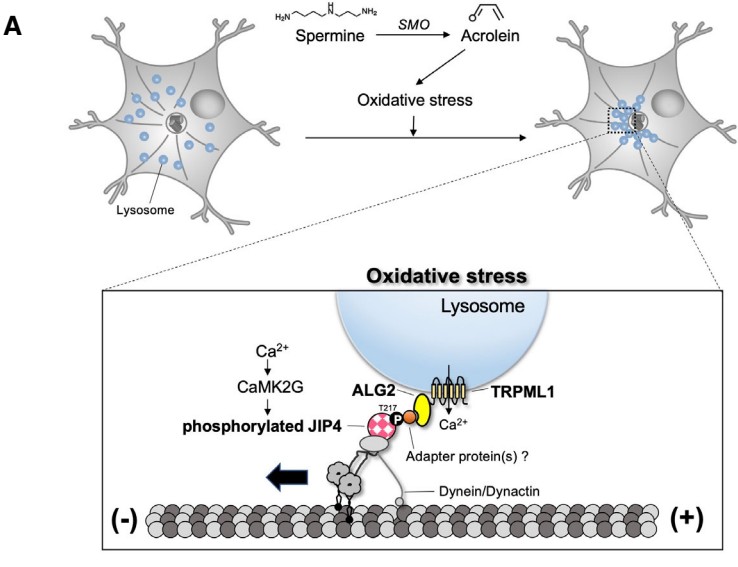

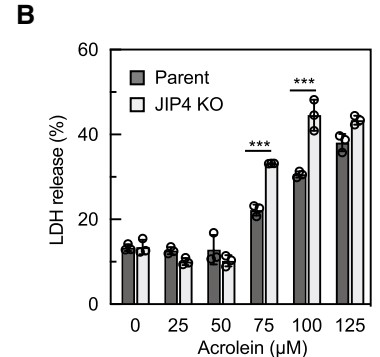 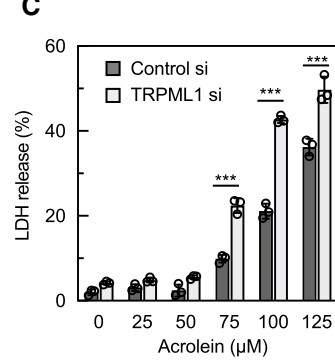

**Figure 7. Schematic illustration of the lysosomal retrograde transport in response to oxidative stress.**

A  Proposed lysosomal retrograde transport signalling in response to oxidative stress. Oxidative stress induces Ca²⁺ release from TRPML1, resulting in JIP4 T217 phosphorylation by CaMK2G. Phosphorylated JIP4 complexes with ALG2 and TRPML1, thereby promoting lysosomal retrograde transport.

B  Parental and JIP4 KO SH-SY5Y cells were treated with various concentrations of acrolein for 24 h. Cytotoxicity was measured by an LDH assay. ***$P < 0.0001$ (Tukey–Kramer's test); $n = 3$ from independent cultures. Values are mean ± SD. At least three experiments were replicated.

C  SH-SY5Y cells were transfected with TRPML1 siRNAs for 48 h and then treated with various concentrations of acrolein for 24 h. Cytotoxicity was measured by an LDH assay. ***$P < 0.0001$ (Tukey–Kramer's test); $n = 3$ from independent cultures. Values are mean ± SD. At least three experiments were replicated.

Source data are available online for this figure.

# Materials and Methods

### Ethics statement

The study protocol complied with the Declaration of Helsinki and was approved by the Ethics Committee of Juntendo University (#2012157). Written informed consent was obtained from all participants.

### Enzyme-linked immunosorbent assay

Blood samples were collected at Juntendo University Hospital between 2018 and 2020. Plasma was extracted by a previously described method (Saiki *et al*, 2017). The amount of acrolein adduct was determined by an Acrolein-Lysine Adduct Competitive EIA Kit

(TaKaRa Bio, Otsu, Japan). Sera from healthy control ($n = 22$) and PD patients [$n = 94$; H&Y stage I ($n = 18$), stage II ($n = 42$) stage III ($n = 23$), stage IV ($n = 7$) and stage V ($n = 4$)] was diluted at 1:10 and subjected to the ELISA in accordance with the manufacturer's instruction.

### Materials

Spermine, *N*1-acetylspermidine, *N*8-acetylspermidine, bafilomycin A1, nocodazole, BAPTA-AM and YM201636 were purchased from Sigma-Aldrich (St. Louis, MO). *N*1-Acetylspermine, *N*1, *N*12-diacetylspermine, *N*1, *N*8-diacetylspermidine, acrolein and hydrogen peroxide were purchased from FUJIFILM Wako Pure Chemical (Osaka, Japan). Aminoguanidine hydrochloride was from Tokyo Chemical Industry Co., Ltd. (Tokyo, Japan). Ciliobrevin D, Jak3

inhibitor VI and Gö 6976 were from Merck Millipore (Burlington, MA).

## Cell culture

SH-SY5Y cells (ATCC #CRL-2266) were cultured in DMEM supplemented with 10% foetal bovine serum, 100 U/ml penicillin/streptomycin (Nacalai Tesque, Kyoto, Japan), MEM non-essential amino acid solution (Thermo Fisher Scientific, Waltham, MA), 1 mM sodium pyruvate and 2 mM L-glutamine at 37°C with 5% $CO_2$. For starvation treatment, cells were washed twice with PBS and cultured in amino acid-free DMEM (048-33575; Wako) without FBS. GFP-LC3 (Kabeya *et al*, 2000), JIP4-myc-DDK (OriGene Technologies, Inc. Rockville, MD; #RC226382) and TRPML1-mCherry stably expressing SH-SY5Y cells were established by transfection of the respective vectors into SH-SY5Y cells using Lipofectamine LTX (Thermo Fisher Scientific), followed by selection with G418 (Roche Diagnostics, Indianapolis, IN) or puromycin (Thermo Fisher Scientific).

## Western blotting

Western blot analysis was performed as previously described with minor modifications (Sasazawa *et al*, 2012, 2015). Cells were lysed in lysis buffer (25 mM Tris–HCl pH 7.6, 150 mM NaCl, 1% NP-40, 1% sodium deoxycholate, 0.1% sodium dodecyl sulphate and protease inhibitor cocktail) for 15 min on ice and centrifuged at 20,000 *g* for 15 min to yield soluble cell lysates. For immunoblotting, 20 μg proteins in cell lysates were subjected to 10–20% gradient SDS–polyacrylamide gel electrophoresis. Proteins were transferred onto a polyvinylidene fluoride (PVDF) membrane and probed with specific antibodies, followed by detection using West Dura Extended Duration Substrate (Thermo Fisher Scientific) and a LAS-4000 mini (GE Healthcare UK Ltd, Buckinghamshire, England). The primary antibodies used were as follows: anti-LC3B (Cell Signaling Technology Inc., Danvers, MA, AB_2137707), anti-β-actin (Merck Millipore, AB_2223041), anti-phospho-p70S6K (Cell Signaling Technology Inc. AB_2269803), anti-p70S6K (Cell Signaling Technology Inc. AB_390722), anti-phospho-S6 (Cell Signaling Technology Inc., AB_916156), anti-S6 (Cell Signaling Technology Inc. AB_331355), anti-α-tubulin antibody (Sigma-Aldrich, AB_477583), anti-ALG2 (R&D Systems, Inc., Minneapolis, MN, AB_10972311), anti-JIP4 (Abcam, AB_299021), anti-His tag (MBL International Co., WOBURN, MA, AB_10597733), anti-GFP (Abcam, AB_303395), anti-RFP (MBL International Co., M204-3) and anti-RUFY3 (Novus Biologicals, Centennial, Co., AB_11022810).

To detect the phosphorylated form of JIP4, lysates were subjected to 6% Phos-tag (50 μmol/l) acrylamide gel (FUJIFILM Wako Pure Chemical) electrophoresis. After electrophoresis, the gel was soaked in 5 mM EDTA, 25 mM Tris and 192 mM glycine for 30 min and then in transfer buffer [25 mM Tris, 192 mM glycine and 10% (vol/vol) methanol] for 10 min. Proteins were transferred to a PVDF membrane and probed with the anti-JIP4 antibody. For dephosphorylation, cells were collected in NEB buffer (50 mM HEPES, 100 mM NaCl, 2 mM DTT and 0.01% Brij 35, pH 7.5) containing 1% Triton X-100 and protease inhibitor cocktail, and treated with lambda phosphatase in the presence of 1 mM $MnCl_2$ for 60 min.

## Subcellular fractionation

Cells were collected in PBS and incubated with 1 mM DSP in PBS for 1 h on ice for cross-linking. The cross-link reaction was stopped by incubation with 10 mM glycine in PBS for 10 min and cells were resuspended in buffer (10 mM Hepes-KOH, 1 mM EDTA, pH 7.4, 250 mM sucrose and phosphatase inhibitor). Cells were disrupted by passing through a 25 G needle 30 times. Debris was removed by centrifugation at 1,000 *g* for 10 min, and the supernatant was centrifuged at 12,500 *g* for 10 min. The resulting supernatant was collected as the lysosome-enriched fraction.

## Immunofluorescence

Cells were fixed with 4% paraformaldehyde (FUJIFILM Wako Pure chemical) for 30 min and permeabilised with 50 μg/ml digitonin in PBS for 15 min. After rinsing three times with PBS, the cells were incubated in blocking buffer (3% bovine serum albumin in PBS) for 30 min and then stained with primary and secondary antibodies at room temperature for 1 h each. Fluorescence images were obtained under an LSM880 confocal laser scanning microscope (Carl Zeiss, Oberkochen, Germany). The primary antibodies used were as follows: anti-LAMP2 (Development Studies Hybridoma Bank, Iowa City, IA; clone H4B4, AB_2134755), anti-cathepsin B (Cell Signaling Technology Inc., AB_2687580), anti-γ-tubulin (abcam, AB_2904198) and anti-JIP4 (Thermo Fisher Scientific., AB_2642850).

## Quantification of lysosomal distribution

To quantify the distribution of lysosomes, Fiji software (Image J. ver.2.1.0/1.53c; Schindelin *et al*, 2012) was used. First, cells were immunostained with anti-LAMP2 and anti-γ-tubulin antibodies for detection of lysosome and MTOC respectively. Z-stack confocal fluorescence images of cells were flattened, and a threshold was applied to eliminate the background. Then, the ROI was drawn as a circle with a radius of 2 μm centred on the spot signal of γ-tubulin. LAMP2 signal intensity in the ROI was measured and the ratio to whole-cell LAMP2 intensity was calculated. At least four images with more than 10 cells per field (more than 40 cells in total) are used for the quantification. The lysosomal distribution value was defined as the ratio relative to control.

## Magic Red-Cathepsin B assay

Cells were grown on glass-bottom 35-mm dishes and treated with chemicals for the appropriate times. Then, cells were incubated with a Magic Red solution for 30 min and then with Hoechst 33342 for an additional 5 min at 37°C. Cells were washed with PBS and then imaged under the LSM880 laser scanning microscope at 37°C with 5% $CO_2$.

## CLEM analysis

Cells were grown on gridded glass-bottom 35 mm dishes. Cells were pre-treated with E64D plus pepstatin A for 1 h and then treated with spm for an additional 4 h. Then, the cells were fixed with 0.1 M phosphate buffer (pH 7.4) containing 100 mM sucrose, 2% paraformaldehyde and 0.2% glutaraldehyde, and observed under

the confocal microscope. Next, cells were refixed with 0.1 M phosphate buffer containing 100 mM sucrose and 2% glutaraldehyde with 1% osmium tetroxide, and then embedded in epoxy resin for electron microscopy. Areas observed by fluorescence microscopy were searched using the grid pattern as a guide.

### siRNA and plasmid transfection

Transfection of SH-SY5Y cells with siRNAs was performed using Lipofectamine RNAiMax (Thermo Fisher Scientific) in accordance with the manufacturer's instructions. The siRNAs were as follows: TRPML1 (SASI_Hs01_00067195); ALG2 (SASI_Hs01_00030269); Rab7a (SASI_Hs01_00104360); JIP4 (SASI_Hs01_00194613); TMEM55A (SASI_Hs02_00352240); TMEM55B(SASI_Hs02_0032 2347); JAK3 (SASI_Hs01_00118128); CaMK2A (SASI_Hs01_ 00189888); CaMK2B (SASI_Hs01_00230999); CaMK2D (SASI_Hs01_ 00202407); CaMK2G (SASI_Hs01_00118118); JIP1 (SASI_Hs01_00 140017); JIP2 (SASI_Hs01_00153848); JIP3 (SASI_Hs01_00050164) and RUFY 3 (#1: SASI_Hs02_00308292; #2: SASI_Hs01_00146952) from Sigma Aldrich; and non-coding siRNA from Dharmacon (Lafayette, CO).

### Quantitative RT–PCR

Total RNA was extracted from cells using an RNeasy mini kit (Qiagen, Venlo, Netherlands) in accordance with the manufacturer's protocol. Reverse transcription was performed using ReverTra Ace qPCR RT Master Mix (TOYOBO, Osaka, Japan). qRT–PCR was carried out using Fast SYBR Green Master Mix and the ABI 7500 Real-Time PCR System (Thermo Fisher Scientific). The following primers were used in this study: human MCOLN1: forward 3′-gagtgggtgcgacaagtttc-5′, reverse 3′-tgttctcttcccggaatgtc-5′; human rab7: forward 3′-gaccaaggaggtgatggtgg-5′, reverse 3′-cacaccgagagactggaacc-5′; human TMEM55B: forward 3′-catttcccccgtttcccga-5′, reverse 3′-tgatcataggggcactccca-5′; human JIP4: forward 3′-tcagccgactttttcagctcc-5′, reverse 3′-acatgagacgtgggtgcatt-5′; human TMEM55A: forward 3′- tgggcactgtggaaacacat-5′, reverse 3′-atctggggtgccaacagtta-5′ and human CaMK2G: forward 3′- gagactgtggagtgtttgcg-5′, reverse 3′-tctgcctgccaactgagaagt-5′.

### Immunoprecipitation

Cells were washed with ice-cold PBS, resuspended in IP Buffer (for detection of TMEM55B-myc-DDK–JIP4 interaction: 50 mM Hepes-KOH, pH 7.5, 300 mM KCl, 1 mM MgCl$_2$, 1 mM EGTA, 10% glycerol and protease and phosphatase inhibitor cocktail with 1% Triton X-100; for detection of TRPML1-mCherry–GFP–ALG2 interaction: 20 mM Tris–HCl, pH 8.0, 150 mM NaCl, 1 mM CaCl$_2$, 10% glycerol and protease and phosphatase inhibitor cocktail with 1% dodecyl maltoside and 0.2% deoxycholic acid) and lysed by passing the samples 10 times through a 25 G needle. Cell lysates were centrifuged at 20,000 g for 10 min at 4°C, and soluble fractions were collected. Cell lysates were incubated with anti-flag M2 (Sigma Aldrich) or anti-RFP magnetic beads (MBL International Co.) for 2 h at 4°C with constant rotation. The beads were washed with each IP buffer with 0.1% Triton X-100 for detection of TMEM55B-myc-DDK–JIP4 interaction and with 0.5% dodecyl maltoside, 0.1% deoxycholic acid for detection of TRPML1-mCherry–GFP-ALG2 four times and eluted with SDS sample buffer.

### Measurement of intracellular free calcium

Cells were washed with balanced salt solution (BSS) buffer (20 mM HEPES-KOH, 135 mM NaCl, 5.4 mM KCl, 2 mM CaCl$_2$ and 10 mM glucose) twice and then incubated with BSS buffer containing 0.01% F-127 and 5 μM Fura2-AM (Dojindo) in the dark for 30 min at 37°C. Data acquisition and analysis were performed using AquaCosmos 2.0 (Hamamatsu Photonics, Hamamatsu, Japan). Solutions were superfused at a rate of 2 ml/min. Intracellular Ca$^{2+}$ levels were determined by the fluorescence ratio ($F_{340}/F_{380}$) at every 5 s interval.

### Sample preparation for MS

JIP4-myc-DDK-overexpressing SH-SY5Y cells treated with 40 μM acrolein with or without 10 μM Jak3 inhibitor VI were lysed with lysis buffer (50 mM Tris–HCl, pH 7.4, 150 mM NaCl, 1% Triton X-100, protease inhibitor cocktail and phosphatase inhibitor cocktail). Cell lysates were reacted with anti-flag M2 magnetic beads (Sigma-Aldrich) for 2 h at 4°C, and then washed with lysis buffer five times. Then, JIP4 protein was eluted with 1 mg/ml flag peptide/PBS.

The sample reduction was performed for 15 min at 56°C with 10 mM DTT. Subsequent alkylation was performed in the dark for 30 min at room temperature with 55 mM iodoacetamide. After chloroform–methanol precipitation of protein, the precipitates were resuspended in 8 M urea and 0.1 M Tris–HCl, pH 8.5. Samples were diluted in 4 M urea with 0.1 M Tris–HCl, pH 8.5 and digested for 2 h with Trypsin/Lys-C Mix (Promega), followed by dilution to 1 M urea with 0.1 M Tris–HCl, pH 8.5, overnight at 37°C. After stopping the digestion with 1% formic acid, the peptide mixture was subjected to solid-phase extraction (GL-Tip SDB, GL Science) for desalting, and peptide effluents were subsequently lyophilised. Lyophilised peptides were dissolved in 0.1% formic acid containing 2% acetonitrile.

### Mass spectrometry and data analyses

Mass spectrometric analyses were performed on a TripleTOF 5600+ mass spectrometer coupled to an Ekspert nanoLC 415 system with Ekspert cHipLC (Sciex, Framingham, MA). To create a library by information-dependent acquisition, the samples were injected onto a Trap column of 200 μm × 0.5 mm (ChromXP C18-CL) with a 120 Å pore size and 3 μm diameter particles, and 75 μm × 150 mm ChromXP C18-CL analysis column (Sciex). Samples were run with a 100 min gradient from 2 to 40% solvent B (solvent A: 0.1% formic acid; solvent B: 0.1% formic acid and 80% acetonitrile) at a flow rate of 300 nl/min.

The analysis results were searched against the UniProt database (release 2020_02) using ProteinPilot 5.0.2. software (Sciex) with a cut-off of a 1% false discovery rate.

Similar LC gradient and mass spectrometer settings were used in SWATH acquisition mode. The precursor isolation windows overlapped by 1 *m/z*, and cover range was 400–1,250 *m/z*. The variable isolation windows were determined with SWATH variable window calculator tool version 1.0 (Sciex). MS2 spectra were recorded with an accumulation time of 40 ms and covered 100–1,600 *m/z*. Each peptide area was calculated using PeakView 2.2 (Sciex) for quantitative analysis. The list of proteins and their extracted peak areas were

exported to MarkerView 1.3 (Sciex) for further analysis. The peak areas of the phosphopeptides derived from JIP4 were normalised by the total peak areas of the unmodified three most intense peptides of JIP4.

### In vitro kinase assay

100–300 aa of the human JIP4 was amplified and subcloned into pRSET-C vector (Thermo Fisher Scientific). His6-tagged JIP4 (100–300 aa) was expressed in the E. coli BL21 strain, and purified using Capturem His-tagged Purification System (TaKaRa). Two hundred nanogram of His$_6$-tagged JIP4 (100–300 aa) was incubated in 200 ng of CaMK2G (Thermo Fisher Scientific), 20 mM Hepes-KOH, pH7.4, 5 mM MgCl$_2$, 1 mM DTT, 1 mM ATP, 2.5 mM CaCl$_2$ and 10 μg/ml calmodulin from bovine brain (Wako) at 30°C for 1 h. The reaction was quenched by boiling the sample with 4× SDS sample buffer. Phosphorylation signal was detected by Phostag-PAGE (Wako).

### Lyso-IP

TRPML1-mCherry stably expressing SH-SY5Y cells were quickly rinsed twice with PBS and then scraped in KPBS (136 mM KCl, 10 mM KH$_2$PO$_4$ and pH 7.4 was adjusted with KOH) and centrifuged at 1,000 × g for 2 min at 4°C. Pelleted cells were resuspended in KPBS and were disrupted by passing through a 25 G needle with 20 strokes. The homogenate was then centrifuged at 1,000 × g for 2 min at 4°C and the supernatant containing the cellular organelles including lysosomes was incubated with KPBS pre-washed anti-RFP magnetic beads (MBL International Co.) on a rotator at 4°C 30 min. Reacted beads were washed with KPBS, and immunoprecipitated lysosomal fraction was then eluted by boiling with 4× SDS sample buffer for 5 min.

### PLA

SH-SY5Y cells were transfected with the pEGFP-C1 or pEGFP-ALG2 vector and treated with acrolein for 2 h. Cells were fixed, and the PLA was performed using Duolink In Situ Detection Reagents FarRed (Sigma-Aldrich) in accordance with the manufacturer's instructions. Anti-GFP (mouse; Proteintech, AB_11182611) and anti-JIP4 (rabbit, Thermo Fisher Scientific, AB_2642850) antibodies were used as primary antibodies. Images were obtained under the LSM880 confocal laser scanning microscope.

### LDH assay

Cytotoxicity of acrolein was measured using a Cytotoxicity LDH Assay Kit-WST (Dojindo, Kumamoto, Japan) in accordance with the manufacturer's procedure. The average absorbance of each triplicate set of wells was calculated, subtracted from the background control value and normalised to the total LDH activity in the cell.

### Statistical analysis

Statistical analysis was performed using JMP Pro (ver. 15.2.0) or Graph Pad Prism (ver. 9.4.1) software. Details of each statistical method are shown in the respective figure legends.

## Data availability

Data supporting the findings of this study are available from the corresponding author upon reasonable request. Source data are available online. MS data have been deposited in ProteomeXchange and jPOST with the accession codes PXD032058 (http://proteomecentral.proteomexchange.org/cgi/GetDataset?ID=PXD032058) and JPST001512 (http://repository.jpostdb.org/entry/JPST001512) respectively.

Expanded View for this article is available online.

## Acknowledgements

We thank Drs. M. Komatsu, H. Morishita, Y. Ichimura (Juntendo University) and Y. Yasueda (JAPAN TOBACCO INC.) for technical advice, and Dr. M. Kitagawa, Dr. K. Sumiyoshi and Ms. Y. Imamichi (Juntendo University) for technical support. This study was supported by Japan Society for the Promotion of Science (JSPS) KAKENHI Grant Numbers 18K15464, 21K07425, 18KK0242, 18KT0027 and 22H02986, and Ministy of Education, Culture, Sports, Science and Technology (MEXT) KAKENHI Grant Number 20H05340, and The Naito Foundation. We also thank the Molecular Profiling Committee, the Grant-in-Aid for Scientific Research on Innovative Areas "Advanced Animal Model Support (AdAMS)" from the MEXT, Japan (KAKENHI 16H06276), for supplying the SCADS Inhibitor Kits. We thank the Research Institute for Diseases of Old Age, the Laboratory of Proteomics and Biomolecular Science, the Laboratory of Molecular and Biochemical Research, and the Laboratory of Cell Biology, Biomedical Research Core Facilities and Juntendo University Graduate School of Medicine for technical assistance. We also thank Mitchell Arico from Edanz (https://jp.edanz.com/ac) for editing a draft of this manuscript.

## Author contributions

**Yukiko Sasazawa:** Conceptualization; resources; data curation; software; formal analysis; supervision; funding acquisition; validation; investigation; visualization; methodology; writing – original draft; project administration; writing – review and editing. **Sanae Souma:** Data curation; investigation. **Norihiko Furuya:** Data curation; formal analysis; validation; methodology. **Yoshiki Miura:** Data curation; software; formal analysis; methodology. **Saiko Kazuno:** Data curation; investigation; methodology. **Soichiro Kakuta:** Data curation; supervision; investigation; methodology. **Ayami Suzuki:** Formal analysis; validation; investigation; methodology. **Ryota Hashimoto:** Data curation; validation; methodology. **Hiroko Hirawake-Mogi:** Data curation; validation; investigation; methodology; writing – review and editing. **Yuki Date:** Conceptualization; data curation; formal analysis; investigation; writing – review and editing. **Masaya Imoto:** Supervision; funding acquisition; validation; methodology. **Takashi Ueno:** Supervision; investigation; methodology. **Tetsushi Kataura:** Data curation; supervision; funding acquisition; validation; methodology. **Viktor I Korolchuk:** Formal analysis; supervision; validation; methodology. **Taiji Tsunemi:** Validation; investigation; visualization; methodology; writing – original draft; writing – review and editing. **Nobutaka Hattori:** Supervision; funding acquisition; writing – original draft; project administration; writing – review and editing. **Shinji Saiki:** Conceptualization; resources; data curation; software; formal analysis; supervision; funding acquisition; validation; investigation; visualization; methodology; writing – original draft; project administration; writing – review and editing.

### Disclosure and competing interests statement

VIK is a Scientific Advisor for Longaevus Technologies. The rest of the authors declare that they have no conflict of interest.

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
