## [Review Process File · The EMBO Journal]

Oxidative stress-induced phosphorylation of JIP4 regulates lysosomal positioning in coordination with TRPML1 and ALG2

Yukiko Sasazawa, Sanae Souma, Norihiko Furuya, Yoshiki Miura, Saiko Kazuno, Soichiro Kakuta, Ayami Suzuki, Ryota Hashimoto, Hiroko Mogi, Yuki Date, Masaya Imoto, Takashi Ueno, Tetsushi Kataura, Viktor Korolchuk, Tsunemi Taiji, Nobutaka Hattori, and Shinji Saiki

DOI: [10.15252/emboj.2022111476](https://doi.org/10.15252/emboj.2022111476)

Corresponding authors: *Shinji Saiki (ssaiki@juntendo.ac.jp)* , *Nobutaka Hattori (nhattori@juntendo.ac.jp)*

Review Timeline:

Submission Date:	19th Apr 22
Editorial Decision:	25th May 22
Revision Received:	23rd Aug 22
Editorial Decision:	16th Sep 22
Revision Received:	18th Sep 22
Accepted:	21st Sep 22

Editor: Ieva Gailite

Transaction Report:

Thank you for submitting your manuscript for consideration by The EMBO Journal. We have now received a full set of referee reports on your manuscript, which are included below for your information.

As you will see from the comments, all reviewers appreciate the quality of the study and find it of interest. However, reviewers 1 and 3 also indicate a number of open questions that would have to be addressed and clarified before they can support publication of the manuscript. From my side, I find these points reasonable and, based on these positive assessments, I would like to invite you to address the concerns raised by the reviewers in a revised manuscript.

We generally allow three months as standard revision time. As a matter of policy, competing manuscripts published during this period will not negatively impact on our assessment of the conceptual advance presented by your study. However, please contact me as soon as possible upon publication of any related work to discuss the appropriate course of action. Should you foresee a problem in meeting this three-month deadline, please let us know in advance and we may be able to grant an extension.

When preparing your letter of response to the referees' comments, please bear in mind that this will form part of the Review Process File and will therefore be available online to the community. For more details on our Transparent Editorial Process, please visit our website: <https://www.embopress.org/page/journal/14602075/authorguide#transparentprocess>. Please also see the attached instructions for further guidelines on preparation of the revised manuscript.

Please feel free to contact me if you have any further questions regarding the revision. I would be happy to discuss the revision in more detail via email or phone/videoconferencing.

Thank you for the opportunity to consider your work for publication. I look forward to receiving the revised manuscript.

Referee #1:

This study identifies the TRPML1-ALG2-JIP4-dynein axis as a novel contributor to the regulation of lysosomal retrograde transport. The authors show that acrolein, an aldehyde frequently found elevated in Parkinson's disease patients, as well as other polyamines, such as spermine and spermidine, cause clustering of lysosomes at the cell center, and this transport can be blocked by depletion of JIP4, TRPML1 or ALG2. The study presents some novel and interesting information, including the stress-dependent participation of different lysosomal proteins (TMEM55B in the case of starvation and TRPML1 for acrolein-induced oxidative stress) linking lysosomes to the JIP4-dynein complex; as well as the importance of CaMK2G-induced JIP4

phosphorylation at T217 for lysosomal cargo selection. However, there are a number of additional experiments that should be performed in order to consolidate the main conclusions. Here are some comments to improve this manuscript.

Major concerns:

1. It is a little surprising that the authors failed to mention and address two recent papers from the Bonifacino (PMID: 35314674) and Tuli (PMID: 35314681) laboratories demonstrating the participation of RYFY3 and RUFY4 in lysosomal retrograde transport. In particular, the Tuli lab described a role of JIP4 linking Arl8b and RUFY3 to the dynein complex. It would be important to assess whether depletion of RUFY3 prevents acrolein-induced lysosomal clustering.
2. Whilst the changes in lysosomal distribution showed by immunofluorescence are clear, the images just show a few cells. It is important to quantify these changes, at least for some critical experiments, such as those shown in Figures 2a, 3d, 6d and 6f, and preferably by measuring changes in cumulative intensity distribution as described by Starling et al., 2016 (PMID: 27113757).
3. The authors should further discuss their interpretation on why JIP4 needs to bind different lysosomal proteins in order to promote lysosomal clustering under different stress conditions. After all, TFEB-mediated upregulation of both TMEM55B and TRPML1 has been described under starvation and oxidative stress conditions. Could TMEM55B and TRPML1 localize to different lysosomal populations and if so, could this be visualized by electron microscopy?
4. A caveat of this study is that it fails to show a phosphorylation-dependent interaction between JIP4 and the TRPML1-ALG2 complex by immunoprecipitation, even when using recombinant proteins. Furthermore, the co-localization between JIP4 and ALG2 shown in Figure 5d is not very convincing and there is not a clear clustering of GFP-ALG2 puncta in response to acrolein treatment as one would expect if the compound induced calcium-mediated binding of ALG2 to TRPML1. It is also important to consider that ALG2 also binds Sec31A and under over-expression localizes to ER exit sites (PMID: 16957052), suggesting that the puncta shown in Figures 5d and 5e may not correspond to lysosomes.
5. The authors should check whether acrolein-induced lysosomal retrograde transport is inhibited by treatment with antioxidants such as N-acetyl-L-cysteine (NAC).
6. The interpretation of this study is that acrolein-mediated lysosomal retrograde transport results in mTORC1 inactivation, and this inactivation then contributes, at least in part, to autophagy induction. However, another possibility is that the observed mTORC1 inactivation is independent of lysosomal trafficking. The authors should measure mTORC1 activity in cells treated with acrolein + nocodazole (or ciliobrevin D) vs nocodazole alone.
7. Does TRPML1 depletion affect lysosomal clustering in response to starvation? Include these data in Figure 6f and add quantification and statistical analysis for all the conditions.
8. The authors do not show that JIP4 is a direct CaMK2G target or whether CaMK2G affects TRPML1-mediated calcium release in response to acrolein.

Minor comments:

1. Figure 1 seems a little bit disconnected from the rest of the study as the authors never assess whether acrolein actually has a positive (by enhancing autophagy) or negative (high toxicity) effect on the pathophysiology of PD.
2. In Figures 6h and S7c, also include the effect of increasing concentrations of acrolein on viability in cells depleted of either TMEM55B or TRPML1.
3. It is unclear why TMEM55B depletion prevents to some extent lysosomal clustering in response to acrolein. Could TMEM55B and TMEM55A show some redundancy under these experimental conditions? Does simultaneous depletion of both proteins prevent lysosomal transport to the cell center in acrolein-treated cells?

Referee #2:

The study by Sasazawa and collaborators begins with the observation that the toxic metabolite acrolein (a product of spermidine metabolism) is elevated in the serum of Parkinson's disease patients. Studies in SH-SY5Y cells showed that acrolein induces perinuclear clustering of lysosomes as well as autophagy. Clustering occurred around the microtubule-organizing center, was abolished by inhibition of microtubule polymerization and dynein motors, and was not accompanied by lysosomal damage or lysosomal enzyme inhibition. Subsequent experiments showed that acrolein inhibits the activity of the autophagy inhibitor mTORC1, thereby increasing autophagic flux.

Testing of three candidate pathways for dynein-mediated lysosomal retrograde transport led to the identification of the TRPML1-

ALG2 pathway as the one responsible for the observed acrolein-induced effects. Importantly, the authors show that also JIP4 plays an essential role in this pathway independently of its classic partner protein, TMEM55B. Biochemistry experiments and the screening of kinase inhibitor libraries identified CaMK2G as a kinase that (upon TRPML1-mediated calcium release) phosphorylates JIP4 at T217 to trigger lysosomal retrograde movement and autophagy.

Finally, JIP4 was found to be essential also in lysosomal retrograde transport triggered by H2O2 and nutrient starvation, even though additional experiments clarified that two different JIP4-mediated mechanisms are in play: Oxidative stress (acrolein, H2O2) triggers lysosomal retrograde transport via the phospho-JIP4-TRPML1-ALG2 pathway, whereas starvation triggers lysosomal retrograde transport via the TMEM55B-non-phosphorylated JIP4 (T217) pathway. Knockout of JIP4 in SH-SY5Y left the cells more vulnerable to acrolein exposure, indicating that JIP4 is part of a protective stress mechanism that acts through autophagy to counteract acrolein toxicity.

The study is very elegantly executed with a vast array of techniques, experiments mostly organized in a mechanistic sense, and appropriate controls. The presentation of the data is straightforward and the story is well built, referenced, and discussed. The data are novel and important and clarify the mechanisms by which JIP4 regulates lysosomal retrograde transport according to two different pathways, importantly establishing CaMK2G-mediated phosphorylation of JIP4 as the switch between the two pathways. This is a complete and compelling story and I don't have any experimental suggestion for this present manuscript (additional components of these pathways could be explored in subsequent manuscripts). My only observation is that the data in Appendix Figure 7d should be presented in the Results section rather than in the Discussion section.

Referee #3:

Lysosomal retrograde transport or positioning is crucial in stress-induced autophagy activation. In the manuscript, Sasazawa et al show that JIP4 phosphorylation at T217 by CaMK2G is critical for lysosomal positioning/clustering under oxidative stress (e.g., acrolein, H2O2), and TRPML1 and ALG2 also engage with the process though mediating the interaction between the lysosome and the microtubule motor complex. The action of mechanism is vital for autophagy activation under the type of stress. On the other hand, starvation-induced lysosome positioning is independent of JIP4 phosphorylation, but depends on the TMEM55B-JIP4 pathway.

The study is of high interest, as it, for the first time, provides direct evidence showing that JIP4 phosphorylation plays pivotal roles in lysosomal retrograde transport and autophagy activation under specific stress settings, while starvation-induced lysosomal retrograde transport is involved in the parallel TMEM55B-JIP4 pathway. The current study offers mechanistic insight into stress-induced lysosomal positioning and autophagy activation, suggesting a new approach to modulate autophagy by tackling JIP4 phosphorylation. Overall, the findings are novel, the manuscript is well written, experiments are properly designed and controlled. The conclusions are supported by the extensive data presented, and the study is a concrete piece of work.

The reviewer has a few specific comments as appended:

1. In Fig 3g, how was the percentage (cells with spikes) at 0 frequency defined? At frequency 1, no TRPML1 knockdown (hollow) graph bar can be seen.
2. In Fig 4, it is interesting that T217 at JIP4 was identified as the site of phosphorylation by CaMK2G. It should be useful to perform alignment analysis for the protein sequences around T217 of JIP4 from a variety of species, and see if the T217 site and its neighbouring sequences (potential CaMK2G phosphorylation consensus motif) are conserved across different species.
3. In Fig 4i, does the bar level indicate the phosphorylation level of each peptide as shown? Please clarify this.
4. Statistical analysis is indicated in the figures. Please detail statistical analysis in the figure legends or/and methods.
5. In the introduction, the content about autophagy is minimal. The background about autophagy process should be included. This will offer opportunities to introduce autophagosome maturation, which is involved in lysosomal transport and fusion. Such the content is relevant to the studies.
6. In Fig 1e and Fig S3, acrolein, Spd or Spm markedly reduces p70S6K phosphorylation marking mTOR activity. Does inhibition of mTOR activity contribute to JIP4 phosphorylation and lysosomal positioning, or does lysosomal positioning regulate mTOR activity, or both could be the case? It is worth discussing this in the discussion section.

Dear Dr. Ieva Gailite,

Thank you for the review of our paper entitled "Phosphorylation of JIP4 regulates lysosomal positioning in coordination with TRPML1 and ALG2" (EMBOJ-2022-111476) and giving us the opportunity to revise the manuscript. We have carefully read the critiques and performed extensive revision experiments. Below we include point-by-point responses to the questions raised by the reviewers. We have also highlighted the revised sentences in the manuscript in red. The numbers of pages and lines are indicated based on the MS Word revised manuscript. We believe this revision appropriately addresses the issues raised by Reviewers. Finally, all the authors would like to thank again the Editor and Reviewers for improving our manuscript by providing their invaluable comments and suggestions.

Referee #1:

This study identifies the TRPML1-ALG2-JIP4-dynein axis as a novel contributor to the regulation of lysosomal retrograde transport. The authors show that acrolein, an aldehyde frequently found elevated in Parkinson's disease patients, as well as other polyamines, such as spermine and spermidine, cause clustering of lysosomes at the cell center, and this transport can be blocked by depletion of JIP4, TRPML1 or ALG2. The study presents some novel and interesting information, including the stress-dependent participation of different lysosomal proteins (TMEM55B in the case of starvation and TRPML1 for acrolein-induced oxidative stress) linking lysosomes to the JIP4-dynein complex; as well as the importance of CaMK2G-induced JIP4 phosphorylation at T217 for lysosomal cargo selection. However, there are a number of additional experiments that should be performed in order to consolidate the main conclusions. Here are some comments to improve this manuscript.

Major concerns:

1. It is a little surprising that the authors failed to mention and address two recent papers from the Bonifacino (PMID: 35314674) and Tuli (PMID: 35314681) laboratories demonstrating the participation of RYFY3 and RUFY4 in lysosomal retrograde transport. In particular, the Tuli lab described a role of JIP4 linking Arl8b and RUFY3 to the dynein complex. It would be important to assess whether depletion of RUFY3 prevents acrolein-induced lysosomal clustering.

(RESPONSE)

As per Reviewer's comment, we had investigated our findings in light of the two excellent papers on RUFY3/4. Specifically, we investigated whether depletion of RUFY3 prevents acrolein-induced lysosomal clustering. We prepared 3 different sequences of siRNA (#3 is the same as in PMID: 35314681). We first checked their knockdown efficiency in SH-SY5Y cells by using RUFY3 antibody (Novus Biologicals, #NBP1-89614, same as in PMID: 35314681), and found that all of them could suppress RUFY3 protein expression as shown below. However, in our hands siRNA #3 showed high cytotoxicity in SH-SY5Y cells and thus we assessed lysosomal distribution by using siRNA #1 and #2. As shown in Fig. EV5b, acrolein induced-lysosomal clustering was not suppressed by RUFY3 depletion by either siRNA. We have added related sentences in the introduction (page 5, line 6 to line 8) and discussion sections (page 22, line 18 to page 23, line 6) and provided data in Fig. EV5b.

2. Whilst the changes in lysosomal distribution showed by immunofluorescence are clear, the images just show a few cells. It is important to quantify these changes, at least for some critical experiments, such as those shown in Figures 2a, 3d, 6d and 6f, and preferably by measuring changes in cumulative intensity distribution as described by Starling et al., 2016 (PMID: 27113757).

(RESPONSE)

According to the Reviewer's comment, we have attempted to quantify lysosomal distribution as described by Starling et al., 2016 (PMID: 27113757). Unfortunately, we were unable to apply the suggested protocol to SH-SY5Y cells, which have relatively large nuclei to the whole cytoplasm. Therefore, we have established a novel method to evaluate lysosomal clustering by calculating the ratio of lysosomes existing in the vicinity of MTOC relative to the total. We have re-stained cells with lysosomal marker LAMP2 and MTOC marker γ -tubulin and analysed the lysosomal clustering using FIJI software. Based on the additional experiments, we reconfirmed the effects of acrolein on the lysosomal distribution changes and added these data in the Results section (Fig. 2a, 3d, 6d and 6f) and methods in Materials and Methods section (page 28, line 12 to page 29, line 3).

3. The authors should further discuss their interpretation on why JIP4 needs to bind different lysosomal proteins in order to promote lysosomal clustering under different stress conditions. After all, TFEB-mediated upregulation of both TMEM55B and TRPML1 has been described under starvation and oxidative stress conditions. Could TMEM55B and TRPML1 localize to different lysosomal populations and if so, could this be visualized by electron microscopy?

(RESPONSE)

First, we assessed mRNA expression levels of TMEM55B and TRPML1 in response to acrolein and H₂O₂ treatment and starvation conditions in SH-SY5Y cells and showed no significant changes by each condition as shown in Appendix Fig. 7a. These data indicated that the stress-dependent changes in JIP4-interacting proteins is not due to the alternation of mRNA expression.

Next, to examine the localization of TMEM55B and TRPML1 by electron microscopy, we have purchased all available antibodies as shown below table and checked their reactivity. Unfortunately, only one TMEM55B antibody purchased from Proteintech could detect target protein by immunocytochemistry, but others could not. Therefore, we abandoned the electron microscopy analysis using TRPML1 and TMEM55B antibodies and thus we evaluated whether TMEM55B localized in the lysosomes by immunostaining using super-resolution microscopy. As shown below, endogenous TMEM55B exhibited a patchy and clustered distribution in all LAMP2-positive lysosomes consistent with a previous report (*J Cell Sci* 135: jcs258566, 2022). Although endogenous TRPML1 could not be observed, we could obtain the data showing that TMEM55B distributed on the lysosomes homogeneously, implying no association of TMEM55B distribution with the selection of the two pathways. As we described in discussion section, we propose that phosphorylation of JIP4 may alter the affinity of its binding partners (page 21, line 16 to page 22, line 2).

Antibody list used in this experiment.

		Manufacture	Cat No.	Source	Reactivity
1	TRPML1 antibody	Sigma	SAB1407780	mouse	No
2		Sigma	M8072-200UL	mouse	No
3	TMEM55B antibody	Proteintech	23992-1-AP	rabbit	Yes
4		Sigma	HPA048528-25U L	rabbit	No
5		Abcam	ab129400	rabbit	No

Immunofluorescence analysis using TMEM55B and LAMP2 antibodies.

4. A caveat of this study is that it fails to show a phosphorylation-dependent interaction between JIP4 and the TRPML1-ALG2 complex by immunoprecipitation, even when using recombinant proteins. Furthermore, the co-localization between JIP4 and ALG2 shown in Figure 5d is not very convincing and there is not a clear clustering of GFP-ALG2 puncta in response to acrolein treatment as one would expect if the compound induced calcium-mediated binding of ALG2 to TRPML1. It is also important to consider that ALG2 also binds Sec31A and under over-expression localizes to ER exit sites (PMID: 16957052), suggesting that the puncta shown in Figures 5d and 5e may not correspond to lysosomes.

(RESPONSE)

We have tried to detect JIP4 and the TRPML1-ALG2 complex by immunoprecipitation. GFP or GFP-ALG2 transfected TRPML1-mCherry stably expressing cells were treated with acrolein for 2h, lysed and immunoprecipitated with anti-RFP magnetic beads, and then co-immunoprecipitated JIP4 and GFP-ALG2 were detected. As shown below, co-immunoprecipitated GFP-ALG2 was detected only in conditions of acrolein treatment, indicating that ALG2 interacts with TRPML1 in response to acrolein. However, JIP4 was detected as several smeared bands even though its amount in the IP fraction appeared to be upregulated in response to acrolein treatment. TRPML1 is a six-transmembrane protein. Transmembrane proteins are known to aggregate and be detected as smear in the high molecular region by immunoblotting when the samples were boiled (*PLoS ONE* 15: e0235563, 2020). TRPML1-bound JIP4 may aggregate with TRPML1 by boiling and detect as smear band by immunoblotting same as TRPML1. Or JIP4 may interact indirectly with TRPML1-ALG2 as we discussed in the manuscript. Therefore, we have added only the result of TRPML1-ALG2 interaction in Fig. 5c (page 16, line 16 to line 17).

GFP-ALG2 interacts with TRPML1-mCherry in response to acrolein treatment.

Moreover, we performed Lyso-IP, a method for the rapid isolation of lysosomes (*Science* 358:807-813, 2017), which showed that JIP4 in lysosomal fraction is upregulated by acrolein treatment. We have added the description of these experiments in the Results section (page 16, line 14 to line 16), the Materials and Methods section (page 35, line 6 to line 15) and Fig. EV4. On the other hand, PLA signals showing the ALG2-JIP4 interaction in response to acrolein treatment were not detected as a cluster as shown in Fig 5e. As commented by the Reviewer, ALG2 mainly localizes to cytosol and ER exit sites. A portion of the protein may translocate to lysosomes and control lysosomal distribution. Indeed, GFP-ALG2 only partially co-localizes with lysosomes in response to acrolein as shown in Fig. 5b. JIP4-ALG2 interaction in response to acrolein may occur not only on lysosomes but also in other locations such as ER exit sites, so that PLA signals in Fig 5f showed no clear cluster. Based on these observations, we propose that JIP4-ALG2 complex translocates to lysosome is participate in lysosomal retrograde transport by binding with TRPML1. We have added these conclusions in the Discussion section (page 22, line 8 to line 16).

5. The authors should check whether acrolein-induced lysosomal retrograde transport is inhibited by treatment with antioxidants such as N-acetyl-L-cysteine (NAC).

(RESPONSE)

According to the Reviewer's comment, we have investigated effect of NAC on acrolein-induced lysosomal clustering and found that it indeed inhibits acrolein-induced

lysosomal clustering. We added the description of these experiments in the Results section in association with Fig. EV3c (page 12, line 18 to page 13, line 2).

6. The interpretation of this study is that acrolein-mediated lysosomal retrograde transport results in mTORC1 inactivation, and this inactivation then contributes, at least in part, to autophagy induction. However, another possibility is that the observed mTORC1 inactivation is independent of lysosomal trafficking. The authors should measure mTORC1 activity in cells treated with acrolein + nocodazole (or ciliobrevin D) vs nocodazole alone.

(RESPONSE)

We have examined effect of nocodazole on acrolein-induced mTORC1 inactivation. As shown in Fig. EV2b, nocodazole did not affect the acrolein-suppressed phosphorylation levels of p70S6K and S6, indicating that acrolein induces lysosomal clustering and inactivates mTORC1 independently. These data suggested that acrolein promotes autophagy in the dual effects: the mTORC1 inhibition contributes to upregulation of autophagosome synthesis, and lysosomal clustering enhances the autophagosome-lysosomal fusion. We added related sentences in the Results section (page 9, line 6 to line 10 and page 10, line 3 to 7) and the data in Fig. EV2b.

7. Does TRPML1 depletion affect lysosomal clustering in response to starvation? Include these data in Figure 6f and add quantification and statistical analysis for all the conditions.

We have checked the effect of TRPML1 depletion on starvation-induced lysosomal clustering. As shown in the revised Fig 6f, either siRNA knockdown against TMEM55B or TRPML1 suppressed starvation-induced lysosomal clustering. We replaced Fig 6f and Fig 7a to new ones and added description of these experiments in the Results section (page 18, line 11 to line 13 and line 16 to 17).

In this study, we showed that starvation did not induce lysosomal clustering in JIP4 KO cells, suggesting that JIP4 is an essential factor for starvation-induced lysosomal clustering, whilst the phosphorylated JIP4 was not involved under starvation conditions. These observations imply that starvation-induced lysosomal clustering is controlled by TRPML1-ALG2 and TMEM55B-JIP4 complexes, both of which involve non-phosphorylated JIP4. On the other hand, we identified in this study that oxidative stress-induced lysosomal clustering is mainly controlled by TRPML1-ALG2 pathway that

was regulated by phosphorylated JIP4. Perhaps the adaptor molecule connecting JIP4 and TRPML1-ALG2 may be different between oxidative stress and starvation stress. One possible adaptor candidate in starvation condition may be RUFY3 because RUFY3 KD did not affect acrolein-induced lysosomal clustering in our experiments but suppressed starvation-induced lysosomal clustering as reported by others (*Nat Commun*, 13:1506, 2022; *Nat Commun*, 13:1540, 2022). Although the precise molecular mechanisms of lysosomal retrograde transport in response to starvation remains to be elucidated further, at least our study has identified the indispensable role of phosphorylated JIP4 (T217) in the oxidative stress conditions.

So far, several dynein adaptors have been reported such as the septin protein SEPT9 (*J Cell Biol*, 220: e202005219, 2021), and the protein SNAPIN (*Neuron*, 68:73-86, 2010) in addition to RUFY3/4 and proteins in three main pathways discussed in this paper. The multiple combinations of these adaptors and regulators may allow lysosomes with different functional properties to respond to various physiological stresses. This may allow binding of many dynein molecules to lysosome to win the tug-of-war against kinesin which exerts the equivalent of four to eight dynein-dynactin complexes (*PNAS*, 106:19381, 2009). Further studies are needed to elucidate such complicated mechanism of dynein-dependent lysosomal regrade transport. We added these considerations to the Discussion section (page 22, line 18-page 23 line 12).

8. The authors do not show that JIP4 is a direct CaMK2G target or whether CaMK2G affects TRPML1-mediated calcium release in response to acrolein.

(RESPONSE)

To investigate whether CaMK2G directly phosphorylates JIP4, we have performed an *in vitro* kinase assay. His₆-tagged recombinant truncated JIP4 (100-300aa) was purified from *E. coli* and reacted with recombinant CaMK2G with/without Ca²⁺ and calmodulin. As shown in Fig. 4j, CaMK2G strongly phosphorylated JIP4 (100-300aa) in a Ca²⁺/Calmodulin-dependent manner, which was clearly suppressed by Jak3 inhibitor VI. These data strongly indicated that acrolein phosphorylates JIP4 by CaMK2G directly via Ca²⁺ influx. We have added the description of these experiments in the Results (page 15, line 12 to line 17), Materials and Methods (page 34 line 15 to Page 35 line 4) and added these data in Fig. 4j and 4k.

Next, we have investigated the effect of CAMK2G knockdown with siRNA on the whole cell calcium flux. As shown in Appendix Fig. 4d, slight decrease in Ca²⁺ spike by the knockdown against CAMK2G was detected, but it was not statistically significant, indicating

that CaMK2G acts downstream of TRPML1. Thus, we added these data in the Results section (page 14, line 14 to line 15) and Appendix Fig. 4d.

Minor comments:

1. *Figure 1 seems a little bit disconnected from the rest of the study as the authors never assess whether acrolein actually has a positive (by enhancing autophagy) or negative (high toxicity) effect on the pathophysiology of PD.*

We believe that the marked increase of the serum acrolein levels during any stage of PD provides a strong rationale for mechanistic studies of the acrolein effect on cells. We completely agree with the Reviewer that equivocal interpretation, protective or toxic to the human, is indeed still required. However, based on the cell-based experiments of the study, we can conclude that acrolein might have partial beneficial effect via autophagy enhancement and we would like to highlight this potential role in the disease. Therefore, we would prefer to keep the PD data which suggests that our studies could have relevance to human pathology.

2. *In Figures 6h and S7c, also include the effect of increasing concentrations of acrolein on viability in cells depleted of either TMEM55B or TRPML1.*

We have examined the cytotoxicity of acrolein in TMEM55B- or TRPML1-knockdown cells. Both TMEM55B- and TRPML1-KD resulted in increased vulnerability to acrolein treatment compared to control KD as well as JIP4 KO cells. Depletion of TMEM55B is known to induce lysosomal stress (*Genes Cells* 23:418-434, 2018), which may prevent accurate assessment of acrolein-induced cytotoxicity. For this reason, we would like to include only the results using TRPML1 siRNA to avoid confusion in the Results section. We added these data in Fig.7c (page 19, line 2).

The effect of TRPML1 and TMEM55B-knockdown on acrolein-induced cytotoxicity.

3. It is unclear why TMEM55B depletion prevents to some extent lysosomal clustering in response to acrolein. Could TMEM55B and TMEM55A show some redundancy under these experimental conditions? Does simultaneous depletion of both proteins prevent lysosomal transport to the cell center in acrolein-treated cells?

(RESPONSE)

As Reviewer suggested, we investigated the effect of the knockdown of TMEM55A and/or TMEM55B on acrolein-induced lysosomal clustering and showed TMEM55A is not involved in lysosomal clustering. These results are consistent with previous reports showing that TMEM55A lacks JIP4 binding sites and is not involved in lysosomal movement despite a high degree of homology with TMEM55B (*Nat Commun* 8: 1580, 2017). We have added the description of these experiments in the Results section (page 11, line 2 to line 4) and added these data in Appendix Fig. 3a.

Referee #2:

The study by Sasazawa and collaborators begins with the observation that the toxic metabolite acrolein (a product of spermidine metabolism) is elevated in the serum of Parkinson's disease patients. Studies in SH-SY5Y cells showed that acrolein induces perinuclear clustering of lysosomes as well as autophagy. Clustering occurred around the microtubule-organizing center, was abolished by inhibition of microtubule polymerization and dynein motors, and was not accompanied by lysosomal damage

or lysosomal enzyme inhibition. Subsequent experiments showed that acrolein inhibits the activity of the autophagy inhibitor mTORC1, thereby increasing autophagic flux. Testing of three candidate pathways for dynein-mediated lysosomal retrograde transport led to the identification of the TRPML1-ALG2 pathway as the one responsible for the observed acrolein-induced effects. Importantly, the authors show that also JIP4 plays an essential role in this pathway independently of its classic partner protein, TMEM55B. Biochemistry experiments and the screening of kinase inhibitor libraries identified CaMK2G as a kinase that (upon TRPML1-mediated calcium release) phosphorylates JIP4 at T217 to trigger lysosomal retrograde movement and autophagy.

Finally, JIP4 was found to be essential also in lysosomal retrograde transport triggered by H₂O₂ and nutrient starvation, even though additional experiments clarified that two different JIP4-mediated mechanisms are in play: Oxidative stress (acrolein, H₂O₂) triggers lysosomal retrograde transport via the phospho-JIP4-TRPML1-ALG2 pathway, whereas starvation triggers lysosomal retrograde transport via the TMEM55B-non-phosphorylated JIP4 (T217) pathway. Knockout of JIP4 in SH-SY5Y left the cells more vulnerable to acrolein exposure, indicating that JIP4 is part of a protective stress mechanism that acts through autophagy to counteract acrolein toxicity.

The study is very elegantly executed with a vast array of techniques, experiments mostly organized in a mechanistic sense, and appropriate controls. The presentation of the data is straightforward and the story is well built, referenced, and discussed. The data are novel and important and clarify the mechanisms by which JIP4 regulates lysosomal retrograde transport according to two different pathways, importantly establishing CaMK2G-mediated phosphorylation of JIP4 as the switch between the two pathways. This is a complete and compelling story and I don't have any experimental suggestion for this present manuscript (additional components of these pathways could be explored in subsequent manuscripts). My only observation is that the data in Appendix Figure 7d should be presented in the Results section rather than in the Discussion section.

>> >> Thank you so much for your kind comments on our study. According to your comment we moved Fig. 7d to Results section at Appendix Fig. S4a.

Referee #3:

Lysosomal retrograde transport or positioning is crucial in stress-induced autophagy activation. In the manuscript, Sasazawa et al show that JIP4 phosphorylation at T217 by CaMK2G is critical for lysosomal positioning/clustering under oxidative stress (e.g., acrolein, H₂O₂), and TRPL1 and ALG2 also engage with the process through mediating the interaction between the lysosome and the microtubule motor complex. The action of mechanism is vital for autophagy activation under the type of stress. On the other hand, starvation-induced lysosome positioning is independent of JIP4 phosphorylation, but depends on the TMEM55B-JIP4 pathway.

The study is of high interest, as it, for the first time, provides direct evidence showing that JIP4 phosphorylation plays pivotal roles in lysosomal retrograde transport and autophagy activation under specific stress settings, while starvation-induced lysosomal retrograde transport is involved in the parallel TMEM55B-JIP4 pathway. The current study offers mechanistic insight into stress-induced lysosomal positioning and autophagy activation, suggesting a new approach to modulate autophagy by tackling JIP4 phosphorylation. Overall, the findings are novel, the manuscript is well written, experiments are properly designed and controlled. The conclusions are supported by the extensive data presented, and the study is a concrete piece of work.

>> >> Thank you so much for your astute comments on our study. Based on your comments and suggestions, we revised texts and figures.

The reviewer has a few specific comments as appended:

1. In Fig 3g, how was the percentage (cells with spikes) at 0 frequency defined? At frequency 1, no TRPML1 knockdown (hollow) graph bar can be seen.

(RESPONSE)

We plotted the fluorescence ratio (F340/F380) values for each of more than 100 cells, and spikes with amplitudes greater than 0.05 were counted for each cell. "0 frequency" means no spikes in the cell for 40 min. For example, the value in "0 frequency" in Fig. 3g represents the percentage of cells without spikes relative to the total cells. We replaced the Fig. 3g with a new one and added a detailed description for Fig. 3g in figure legends (page 41, line 10 to line 12).

2. In Fig 4, it is interesting that T217 at JIP4 was identified as the site of phosphorylation by CaMK2G. It should be useful to perform alignment analysis for the protein sequences around T217 of JIP4 from a variety of species, and see if the T217 site and its neighbouring sequences (potential CaMK2G phosphorylation consensus motif) are conserved across different species.

(RESPONSE)

We have added the alignment information of JIP4 sequence in various species and human JIP1-JIP4 in Fig. EV5a. It demonstrates strong conservation of the phospho-site and adjacent sequences.

3. In Fig 4i, does the bar level indicate the phosphorylation level of each peptide as shown? Please clarify this.

(RESPONSE)

Fig. 4i represents the number of phosphorylated peptides in each condition. We added this clarification to the figure legend for Fig. 4i (page 42 line 15).

4. Statistical analysis is indicated in the figures. Please detail statistical analysis in the figure legends or/and methods.

(RESPONSE)

We have described the details of statistical analyses in the figure legends and the Materials and Methods section (page 36, line 14 to line 16).

5. In the introduction, the content about autophagy is minimal. The background about autophagy process should be included. This will offer opportunities to introduce autophagosome maturation, which is involved in lysosomal transport and fusion. Such the content is relevant to the studies.

(RESPONSE)

We have added the background about autophagy process in the Introduction section as per Reviewer's suggestion (page 4, line 9 to line 16).

6. In Fig 1e and Fig S3, acrolein, Spd or Spm markedly reduces p70S6K

phosphorylation marking mTOR activity. Does inhibition of mTOR activity contribute to JIP4 phosphorylation and lysosomal positioning, or does lysosomal positioning regulate mTOR activity, or both could be the case? It is worth discussing this in the discussion section.

(RESPONSE)

Starvation rapidly inactivates mTORC1 as shown **below (left)**. On the other hand, starvation did not affect JIP4 phosphorylation as shown in Fig 6e. These data indicate that, inhibition of mTORC1 activity did not contribute to JIP4 phosphorylation. Moreover, Torin-1, a mTOR specific inhibitor, did not show neither JIP4 phosphorylation **(below right)** nor lysosomal distribution change **(Fig. EV2c)**. On the other hand, nocodazole, which inhibits lysosomal movement, did not affect the acrolein-suppressed phosphorylation level of p70S6K and S6 **(Fig. EV2b)**, indicating that acrolein induces lysosomal clustering and inactivates mTORC1, independently. We propose that these two effects of acrolein promote autophagy more effectively, because mTORC1 inactivation contributes to autophagosome upregulation, and lysosomal clustering contributes to autophagosome-lysosomal fusion. We added related sentences in the Results section **(page 9, line 6 to line 10 and page 10, line 3 to 7)** and in **Fig. EV2b,c**.

Figures for reviewers removed

Thank you for submitting a revised version of your manuscript. Your study has now been seen by two of the original referees, who find that their main concerns have been addressed and now recommend publication of the manuscript. I am therefore pleased to inform you that your manuscript has been accepted for publication in The EMBO Journal.

There remain a couple of issues that have to be addressed before I can formally accept your manuscript:

1. Please address the remaining minor comments from reviewer #1.

Please let me know if you have any further questions regarding any of these points. You can use the link below to upload the revised files.

Referee #1:

I have now carefully read the revised manuscript and the point-by-point rebuttal letter. I found that the authors have put a great amount of effort into providing comprehensive and appropriate responses to the reviews' comments and have considerably improved the manuscript by including quantifications and additional controls.

I just have a couple minor comments. Please double check the label of the graphic shown in Figure 6d. I am assuming that the second bar correspond to cells treated only with H₂O₂, while the third and fourth bars represents cells incubated with H₂O₂+BAMPTA-AM and H₂O₂+Jak3 inhibitor VI, respectively. This needs to be corrected.

In addition, the data shown in the rebuttal letter in response to the minor point 2 (Effect of TRPML1 and TMEM55B-knockout on acrolein-induced cytotoxicity) does not correspond to the results shown in Figure 7c. Once again, I am assuming that the legend of the rebuttal letter's figure is not correct and that the bars of a darker color correspond to siTRPML1- instead of siControl-treated cells.

Referee #3:

The authors have now satisfactorily addressed all my previous comments in the revised manuscript. It is now ready for publication.

Response to Reviewer 1:

Referee #1:

I have now carefully read the revised manuscript and the point-by-point rebuttal letter. I found that the authors have put a great amount of effort into providing comprehensive and appropriate responses to the reviews' comments and have considerably improved the manuscript by including quantifications and additional controls.

I just have a couple minor comments. Please double check the label of the graphic shown in Figure 6d. I am assuming that the second bar correspond to cells treated only with H₂O₂, while the third and fourth bars represents cells incubated with H₂O₂+BAMPTA-AM and H₂O₂+Jak3 inhibitor VI, respectively. This needs to be corrected.

>> Thank you for your careful check of the figure. The notation was incorrect and has been corrected.

In addition, the data shown in the rebuttal letter in response to the minor point 2 (Effect of TRPML1 and TMEM55B-knockout on acrolein-induced cytotoxicity) does not correspond to the results shown in Figure 7c. Once again, I am assuming that the legend of the rebuttal letter's figure is not correct and that the bars of a darker color correspond to siTRPML1- instead of siControl-treated cells.

>> We carefully checked both rebuttal letter's figure and Figure 7c. The values of both control siRNA and TRPML1 siRNA in the rebuttal letter are consistent with in Figure 7c. In figure 7c, we excluded the bars of both TMEM55B siRNA and at acrolein 150 μ M from the figure in rebuttal letter. We think that the legend is correct. However, we noticed the spelling error in the figure legend, so we apologize and would like to replace this figure to new one.

Editor accepted the revised manuscript.